# Hybrid Detection Technique for IP Packet Header Modifications Associated with Store-and-Forward Operations

Asmaa Munshi 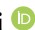

College of Computer Science and Engineering, University of Jeddah, Jeddah 21959, Saudi Arabia;
ammunshi@uj.edu.sa

**Abstract:** The detection technique for IP packet header modifications associated with store-and-forward operation pertains to a methodology or mechanism utilized for the identification and detection of alterations made to packet headers within a network setting that utilizes a store-and-forward operation. The problem that led to employing this technique lies with the fact that previous research studies expected intrusion detection systems (IDSs) to perform everything associated with inspecting the entire network transmission session for detecting any modification. However, in the store-and-forward process, upon arrival at a network node such as a router or switch, a packet is temporarily stored prior to being transmitted to its intended destination. Throughout the duration of storage, IDS operation tasks would not be able to store that packet; however, it is possible that certain adjustments or modifications could be implemented to the packet headers that IDS does not recognize. For this reason, this current research uses a combination of a convolutional neural network and long short-term memory to predict the detection of any modifications associated with the store-and-forward process. The combination of CNN and LSTM suggests a significant improvement in the model's performance with an increase in the number of packets within each flow: on average, 99% detection performance was achieved. This implies that when comprehending the ideal pattern, the model exhibits accurate predictions for modifications in cases where the transmission abruptly increases. This study has made a significant contribution to the identification of IP packet header modifications that are linked to the store-and-forward operation.

**Keywords:** attack surface; attack vectors; network attacks; intrusion detection system; pervasive breaches



## 1. Introduction

In a store-and-forward network, upon packet arrival at a network node such as a router or switch, a packet is temporarily stored prior to being transmitted to its intended destination. Throughout the duration of storage, it is possible that certain adjustments or modifications could be implemented to the packet headers for a variety of reasons, including but not limited to quality of service (QoS) enhancements [1], traffic management, security measures, or routing optimizations. The key research motivation of this present study lie with the fact that computer network systems play a significant role in both the economy and society. The significant increase in data traffic within computer networks can be attributed to several factors. Firstly, the widespread adoption of the Internet has played a crucial role in this growth. Additionally, the expansion of network access to a diverse array of personal devices, including smartphones and cars, has contributed to the surge in data traffic [2]. Lastly, the development of new devices has necessitated a series of reviews to ensure compatibility with primary network protocols [3].

In recent years, there has been an increase in cyber-attacks, particularly against industries and companies that provide online services [4]. Malicious hackers may deploy various types of attacks, such as distributed denial-of-service (DDoS) or the port scan and infiltration attack, to hijack valuable data or make servers unavailable to users. This

requires careful consideration of data security and integrity as companies must assure the data confidentiality, integrity, and availability, which means the data must be appropriately kept, managed, and maintained to prevent illegal access [5]. Several technological solutions have been introduced to reduce the problem, including the use of encryption, authentication systems, antivirus, firewalls, and intrusion detection systems (commonly known as IDSs) [6]. An IDS is a hardware or software system that monitors a company's computer network for potential threats or attacks [7]. Furthermore, IDSs are important for protecting information with growing of unauthorized activities in a network [8]. IDS is a successful technique, but within it, there are many approaches towards which it executes its operation [9]. The artificial neural network (ANN) is an adaptive approach that was utilized by one IDS [10].

While the IDS is successful at continuously monitoring the network for suspicious activity, its operation is too general. There are some silent network operations regarding which the IDS might fall short in performing its operation. This may involve a network where a store-and-forward operation was implemented [11]. In a store-and-forward network, when an IP packet arrives at a network node, such as a router or switch, it is temporarily stored before being transmitted to its intended destination. During the storage phase, in accordance with the network layer protocol convention, the node conducts error-checking on the data packet to verify its integrity and rectify any corrupted data packets [12]. Additionally, it analyzes the destination address within the data packet to ascertain the subsequent hop or node to which the packet should be forwarded. Store-and-forward networks are frequently employed in diverse networking environments, encompassing local area networks (LANs), wide area networks (WANs), and the Internet. The choice of this methodology is contingent upon the particular demands of the network and the nature of the data being conveyed [13].

The research problems identified in this study pertain to the storage phase of the store-and-forward operation. It is customary for the node to perform error-checking on the data packet in accordance with network layer protocol conventions to ensure its integrity and absence of errors. However, a potential issue arises when an adversary interferes and disrupts the packet during this process. Nevertheless, it is conceivable that specific modifications or revisions could be implemented to the packet headers for diverse objectives that could potentially result in packet corruption or spoofing. This suggests that if an adversary has the capability to exploit the given timeframe and carry out an attack, they will have the chance to do so. The traditional intrusion detection system (IDS) does not hinder the store-and-forward operation. Therefore, the potential for a lack of mechanisms to identify occurrences within these operations may arise.

The rationale for formulating this research problem is rooted in the need to mitigate false negatives. The main challenges in detection revolve around generating elements that encompass credible attack signatures or constructing a signature that encompasses all possible permutations of the pertinent attack [14]. The primary focus lies in the significance of detection accuracies, wherein the optimal outcomes are achieved through leveraging the features of the detected elements. This implies that the data input obtained from external sources necessitates adequate training or learning in order to be effectively processed or predicted [14,15]. In light of the aforementioned issues in the store-and-forward context, the present study introduces a novel approach titled "Hybrid IP Packet header modifications detection" to tackle these challenges. The aim of this study is to integrate diverse machine learning prediction techniques in order to effectively detect any modifications that transpire during a store-and-forward operation. This implies that the detection mechanism employs a hybridized approach, incorporating various techniques to detect alterations made to packet headers. One possible approach to resolving the issue at hand could be the incorporation of multiple classifiers into the system.

## 2. Related Work

Extensive study has been conducted in the past on the topics of packet inspection and network attacks. The CICIDS2017 dataset has been widely utilized in the bulk of prior research endeavors. The investigations yielded experimental results that showcased high rates of detection and low rates of false positives. Additionally, the performance of the system was shown to be superior in terms of accuracy, detection rate, false alarm rate, and time overhead, as indicated by previous research [16]. The intrusion detection system (IDS) is the primary area of research for detecting attacks. Machine learning approaches are predominantly utilized for achieving successful detection [17].

The utilization of the CICIDS2017 dataset has been prevalent in the bulk of past research studies. The tests yielded experimental results that showcased high rates of detection and low rates of false positives. Additionally, the performance of the tested methods was shown to be superior in terms of accuracy, detection rate, false alarm rate, and time overhead, as indicated by previous research [18]. The implementation of an intrusion detection system for the controller area network (CAN) bus system in automotive applications has been discussed in a previous study [19]. In the domain of identifying distinct attacks, unsupervised techniques exhibit a higher level of effectiveness when compared to supervised methods [20].

Zeng et al. [21] introduced an approach for intrusion detection system (IDS) that uses a deep learning-based model to effectively detect and classify malicious network traffic aimed at compromising on-board units (OBUs). This methodology obviates the need for human feature extraction and possesses the additional advantage of being capable of processing unprocessed traffic data. The effectiveness of this approach is assessed via a comparison analysis with alternative intrusion detection system (IDS) methods using both a publically accessible dataset and a generated dataset of a vehicular ad hoc network (VANET). The results of the study indicate that the implemented scheme exhibited exceptional performance while necessitating a reduced allocation of resources.

Hidalgo-Espinoza et al. [22] presented a detailed analysis of the procedures employed in the development of an intrusion detection system (IDS) using a deep learning architecture. the primary aim of the proposed system is to ascertain the legality of login attempts conducted on a computer network, effectively differentiating between unauthorized hacking endeavors and allowed actions. Moreover, the authors suggest that additional research should be undertaken, employing a more flexible arrangement of the deep learning framework. Various supervised learning techniques, including support vector machine (SVM), k-nearest neighbor (kNN), random forest, artificial neural networks (ANNs), deep convolutional neural networks (CNNs), and long short-term memory, have been intensively explored in academic research [23].

Given the importance of machine learning in intrusion detection systems (IDSs), it has been observed that the use of supervised learning techniques for training requires a significant amount of labeled data. However, these methods have shown the ability to outperform unsupervised learning techniques in detecting known instances of attacks [24]. This information should contain a diverse range of assault manifestations. Ho et al. [25] conducted a study in which they employed a convolutional neural network (CNN) classifier for the purpose of an intrusion detection system (IDS). A level of accuracy of 99.78% was achieved. In addition, it demonstrates the capacity to discern and detect attacks, a challenge frequently encountered by traditional intrusion detection systems (IDSs).

The study conducted by Choraś [26] demonstrated the methodology of incorporating several hyperparameters and topology configurations in order to attain the highest level of performance for an artificial neural network (ANN) classifier. This was achieved through conducting experiments on a commonly utilized intrusion detection system (IDS) dataset. Additionally, the authors demonstrated the possible influence of hyperparameters on the final outcome of the classification. The influence of a little adjustment in hyperparameter setting on the accuracy of a particular neural network architecture is illustrated by the

authors through the use of two distinct intrusion detection systems (IDSs) as case studies. The best design attains 99.909% accuracy in the task of multi-class categorization.

The reason for this is that unsupervised approaches are specifically designed to detect anomalies within the conventional CAN traffic pattern [27]. The process of creating authentic assault datasets from real autos in the context of in-vehicle networks presents difficulties because to the significant costs associated with it and the necessity to prioritize safety concerns [28]. However, the only requirement for unsupervised learning methods is the collection of data from a vehicle during its normal functioning. The aforementioned data can be easily acquired from a vehicle that is functioning in a standard manner. A considerable proportion of the endeavors focused on detecting intrusions in controller area networks involves the application of deep learning methodologies [29]. Some studies focus on certain components of AIDs, such as temporal factors or payload size, while other study combines these characteristics in a detection model to enhance the identification of a wider range of attacks.

The review of prior research papers on algorithms for detecting IP packet header change demonstrates a wide array of approaches. The efficacy of rule-based systems in detecting established patterns of header modifications linked to store-and-forward operations has been demonstrated. The utilization of machine learning techniques, such as anomaly detection, has been extensively investigated in order to identify minor alterations that are not encompassed by pre-established rules. Moreover, the integration of rule-based and machine learning models in hybrid techniques shows potential in attaining a well-rounded detection capability. It is crucial to note that the aforementioned studies [16–29] continue to prioritize the study of intrusion detection systems (IDSs). However, there has been limited discussion regarding the practical implementation of at-a-point detection in various applications. Therefore, building upon prior effective implementations of intrusion detection systems (IDSs) [4–8] in the field of network security and inspired by the research efforts of [7,19,22], this study introduces methodologies for investigating the application of detection in store-and-forward operations, specifically in the context of transmission sessions. The research of previously published scholarly works on techniques for detecting IP packet header changes demonstrates a wide array of approaches. The usefulness of rule-based systems in identifying known patterns of header alterations associated with store-and-forward operations has been established. The utilization of machine learning techniques, such as anomaly detection, has been extensively investigated in order to identify tiny alterations that are not accounted for using pre-established criteria. Furthermore, the integration of rule-based and machine learning models in hybrid techniques shows potential in attaining a well-rounded detection capability.

The research gap established dwells on the fact that while there have already been significant breakthroughs in the detection algorithms for IP packet header modification within IDS, there is still a discernible research gap in adequately addressing the dynamic and developing characteristics of header alterations associated with store-and-forward processes. Current methodologies frequently encounter difficulties in accommodating novel attack strategies that leverage non-traditional modifications. Moreover, there is a limited body of research that has thoroughly assessed the practical implementation of hybrid models in network environments in real-world scenarios.

## 3. Research Methodology

The study employed a research methodology consisting of two phases. The first phase involved constructing a network scenario to generate a dataset. This dataset was then utilized with a classifier to predict IP packet header modifications associated with store-and-forward operations. The other phase employed the CICIDS2017 dataset and then performed the same prediction of IP packet header modifications that are associated with the store-and-forward operation. Subsequently, the outcome was subjected to a comparative analysis. However, the classifiers were also involved in a hybrid fashion.

### 3.1. Dataset Generation

The dataset for this study is generated from two different approaches: Develop a script capable of extracting the entirety of the IP packet content, with a specific focus on those that exhibit similar features to the CICIDS2017 dataset. The purpose of this research is to establish a network testbed with the goal of initiating a transmission that demonstrates the store-and-forward mechanism. Then, send packet back and forward for many transmissions and capture the transmission for 5 days, similar to the CICIS2017 dataset.

#### 3.1.1. CICIDS2017 Dataset Generation

The CICIDS2017 dataset (see Figure 1) is extensively employed in the field of cybersecurity for the purposes of intrusion detection and network traffic analysis research [17]. The creation and release of the aforementioned project occurred in 2017 under the auspices of the Canadian Institute for Cybersecurity (CIC) at the University of New Brunswick, located in Canada [18]. The primary objective of the dataset is to offer a comprehensive range of network traffic scenarios that accurately reflect real-world conditions. This dataset serves as a means to assess the efficacy of intrusion detection systems (IDSs) and other cybersecurity algorithms in terms of their performance [19].

| Name | Size |
|---|---|
| Friday-WorkingHours-Afternoon-DDos.pcap_ISCX | 75,317 KB |
| Friday-WorkingHours-Afternoon-PortScan.pcap_ISCX | 75,104 KB |
| Friday-WorkingHours-Morning.pcap_ISCX | 56,950 KB |
| Monday-WorkingHours.pcap_ISCX | 172,782 KB |
| Thursday-WorkingHours-Afternoon-Infilteration.pcap_ISCX | 81,155 KB |
| Thursday-WorkingHours-Morning-WebAttacks.pcap_ISCX | 50,804 KB |
| Tuesday-WorkingHours.pcap_ISCX | 131,914 KB |
| Wednesday-workingHours.pcap_ISCX | 219,890 KB |

**Figure 1.** The snapshot of the CICIDS2017 dataset.

The CICIDS2017 dataset contains a total of 78 columns representing various attributes related to transmission sessions, along with an additional column serving as the label. Some of the features of the dataset are presented in Table 1.

**Table 1.** Selected features from CICIDS2017 dataset.

| FD | FP | BP | LF | LB | FL | BL | FF | BB | PL | PP |
|---|---|---|---|---|---|---|---|---|---|---|
| 3 | 2 | 0 | 12 | 0 | 40 | 0 | 666,666.6667 | 0 | 6 | 6 |
| 3 | 2 | 0 | 12 | 0 | 40 | 0 | 666,666.6667 | 0 | 6 | 6 |
| 1022 | 2 | 0 | 12 | 0 | 40 | 0 | 1956.947162 | 0 | 6 | 6 |
| 4 | 2 | 0 | 12 | 0 | 40 | 0 | 500,000 | 0 | 6 | 6 |
| 42 | 1 | 1 | 6 | 6 | 20 | 20 | 23,809.52381 | 23,809.52381 | 6 | 6 |
| 4 | 2 | 0 | 12 | 0 | 40 | 0 | 500,000 | 0 | 6 | 6 |

("Flow Duration [FD]", "Total Forward Packets [FP]", "Total Backward Packets [BP]", "Total Length of Forward [LF] Packets", "Total Length of Backward Packets [LB]", "Forward Header Length [FL]", "Backward Header Length [BL]", "Forward Packets/s [FF]", "Backward Packets/s [BB]", "Minimun Packet Length [PL]", "Maximum Packet Length [PP]").

The dataset encompasses both benign (non-malicious) and malicious network traffic. The category of malicious network traffic encompasses a range of network attacks that are classified under various labels, such as "Bot," "DDoS," "DoS GoldenEye," "DoS

Hulk," "DoS Slow-httptest," "DoS slowloris," "FTP-Patator," "Heartbleed," "Infiltration," "PortScan," "SSH-Patator," "Web Attack—Brute Force," and "Web Attack—SQL Injection". However, it should be noted that the dataset does not directly provide information about the malicious attack itself. Instead, it offers features that can be analyzed and used to infer the presence of an attack. What is the technique inferring or drawing conclusions on based on this characteristic? There exist two distinct methodologies for analyzing attacks in the dataset: manual extraction and automatic deduction.

The manual method encompasses a procedure known as "ground truth labeling". The process of ground truth labeling entails the identification and annotation of individual instances of network traffic, classifying them as either benign or associated with a particular attack category. This is achieved through the utilization of expert knowledge and real-world attack data [30]. The establishment of the validity of the labeling (ground truth) study remains uncertain [31]. This uncertainty arises from the identification of a significant number of previously unrecorded errors throughout the entire process of creating the CIC-IDS 2017 and CSE-CIC-IDS 2018 datasets [31]. These errors encompass various stages such as attack orchestration, feature generation, documentation, and labeling.

The application of an automated labelling process, which utilizes a combination of time and host-based filtering techniques, to effectively segregate malicious traffic has been widely implemented in various scenarios [30]. It was established that two cutting-edge methodologies were usually employed in the automatic approach, namely "Confident Learning" [32] and "O2D-Net" proposed by Huang et al. [33], for the purpose of identifying inaccuracies in labeling within the domain of computer vision.

### 3.1.2. Network Testbed for Dataset Generation

This study successfully established the network scenario, and the script necessitates the simulation of a transmission from one network to another. As depicted in Figure 2, it can be observed that upon entering the transmission session, the initial action taken with the packet is to subject it to a store-and-forward operation. After the completion of the "error checking" process on the store-and-forward mechanism, the packet will undergo an examination of its header information.

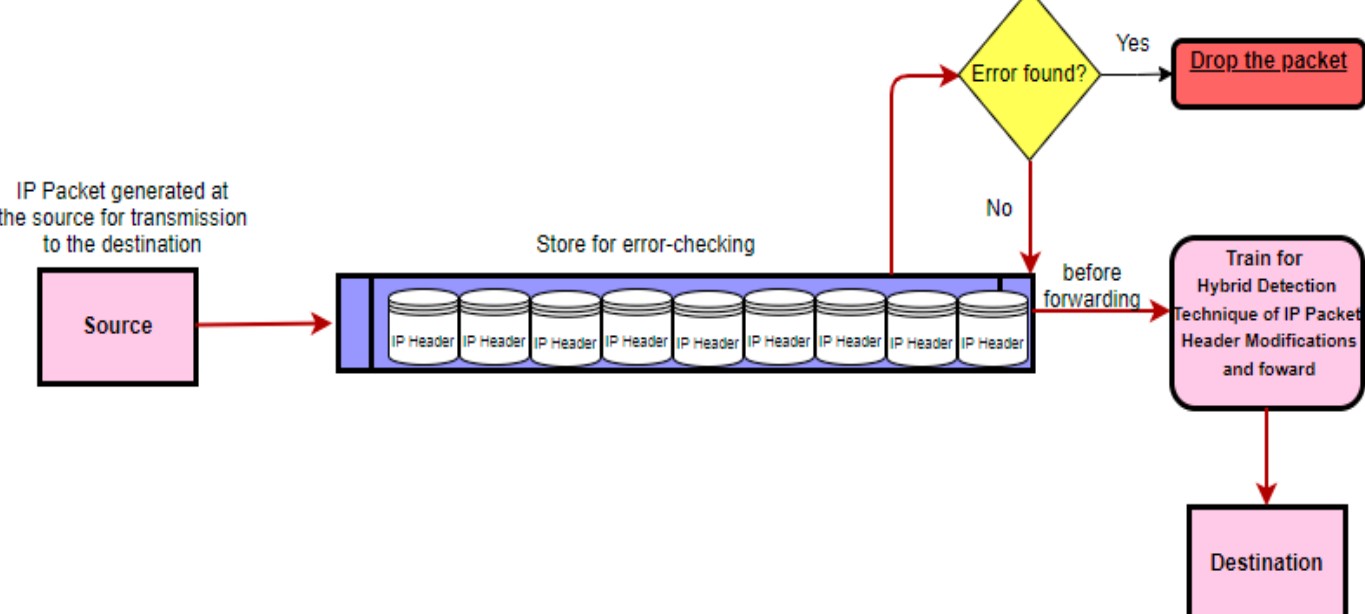

**Figure 2.** A schematics of the network scenario that involves the store-and-forward operation.

Given that the label column of CIC-IDS 2017 dataset entries denotes distinct categories of attacks., the purpose of this study is to investigate the specific alterations made to

the header field in cases where the flow is linked to a certain activity. The analysis of traffic distribution which involves identification of any disparities in the allocation of traffic among servers or services can reveal a DDOS attack. Such disparities may serve as indicators of potential efforts to overload particular resources. Furthermore, since another primary indicator of a DoS attack is "Traffic Volume", this metric can be analyzed in terms of an abnormally elevated influx of incoming network traffic directed towards a particular target or segment of a network. Thus, "Flow Duration" and "Forward Packets/s," can be utilized for the purpose of recording this information. These two features are sufficient to demonstrate the prevalence of DDoS attacks. The features are also present in the CIC-IDS 2017 dataset. Hence, this study aimed to capture and conceptualize these features within a store-and-forward network.

To facilitate the analysis of the packet during the transmission session and generate the dataset, the following steps were performed:

- Scripts that will incorporate a modified value into specific packet header information within designated scenarios were developed.
- An additional script that preserves the default value assigned by the IP protocol in certain packets within the given scenarios was developed.
- An additional script that facilitates the transmission of IP packets recording the default values "Flow Duration" and "forward packets" within the given scenarios, as well as those that incorporate the modified values of the selected IP packet headers within the scenarios, was developed.
- The "Flow Duration" and "forward packets" field information is defined as the set of data elements that can be modified in a given scenario. We composed a dummy value situation wherein a specific value within the "store-and-forward process" was modified.
- We created a script for conducting a comprehensive examination to ascertain the existence of all packet headers.
- An additional script will generate a comprehensive report containing all the information present in the "Flow Duration" and "forward packets" at the specific moment of transmission.
- Subsequently, a script will be developed to construct a binary function that will determine whether the entirety of the IP packet's "Flow Duration"and "forward packets" has been altered or not, represented in a binary set format.
- Another script will be executed iteratively to generate datasets.

The given scenario involves a network router that utilizes the "store-and-forward process" for its operation. The router is responsible for receiving IP packets and temporarily storing them prior to their subsequent transmission to their intended destinations. During the "store" phase, the router is able to conduct a thorough examination and verification of all alterations made to the packet header. In order to comprehensively analyze the situation, it is imperative to delineate the scenario in a sequential manner. Upon the arrival of an IP packet at the router, it is temporarily stored in the router's memory as part of the "store-and-forward process".

During the packet's storage phase, the router conducts an examination of the packet header to detect any alterations made to the number of "Flow Duration" and "forward packets". Packet forwarding occurs after the router has conducted header inspection and made a decision regarding the appropriate forwarding path. At this stage, the router has two options: it can either forward the packet in its original form to the intended destination, or it can undertake any required actions for packets that have been modified. The router has the capability to record the results of its inspection and any actions it has taken in relation to altered packets, with the purpose of conducting subsequent analysis or auditing. This particular scenario holds potential value within the realm of network security, specifically in the areas of intrusion detection and the monitoring of potentially suspicious activities. That is why machine learning is crucial in order to utilize the functions capable of recognizing potential modifications to headers and implementing suitable measures to safeguard the integrity and security of the data being transmitted.

### 3.2. Convolutional Neural Network

The convolutional neural network (CNN) is a highly significant and effective tool in the field of deep learning. It is classified as a type of "feed forward neural network" that is primarily employed in the domains of image processing and pattern recognition [34]. The architecture of a convolutional neural network (CNN) comprises several distinct layers, namely the "input layer", "convolution layer", "pooling layer", and the "output layer". The convolutional layer receives input data and conducts the convolution process through applying a filter to extract a feature map. The pooling layer is responsible for receiving feature maps from the convolution layer and performing down-sampling on these feature maps. During the pooling procedure, a group of n adjacent data points are transformed into a unified format. This is achieved through incorporating a bias term (bx + 1) and a scalar weight (Wx + 1) and applying an activation function. The outcome of this process is the generation of a condensed feature map. One significant benefit of convolutional neural networks (CNNs) lies in their capacity for parallel learning, which contributes to a reduction in network complexity [35]. Enhanced resilience and scalability can be attained through the implementation of the sub-sampling procedure. Equation (1) [35] is a general framework of CNNs and can be used to describe the processing of output at the layers of a CNN:

$$C_{x,y}^{(l,k)} = \tanh\left(\sum_{t=0}^{f-1}\sum_{r=0}^{K_h}\sum_{c=0}^{K_w} W_{(r,c)}^{(k,t)} C_{(x+r,\ x+c)}^{(l-1,t)} + Bias^{(i,k)}\right) \tag{1}$$

$C_{x,y}^{(l,k)}$ is the generated outcome of a neuron at convolution layer $l$, feature pattern $k$, row $x$, and column $y$. $f$ represents the number of convolution cores in a given feature data pattern. At the subsampling stage, the output of neuron at the $lth$ subsampling layer, $kth$ feature pattern, row $x$, and column $y$ is expressed in Equation (2):

$$C_{x,y}^{(l,k)} = \tanh\left(W^{(k)}\sum_{r=0}^{S_h}\sum_{c=0}^{S_w} C_{(x\times S_h+r,\ y\times S_w+c)}^{(l-1,t)} + Bias^{(i,k)}\right) \tag{2}$$

At the $lth$ hidden layer H, the output of neuron $j$ is given in Equation (3):

$$C_{(l,j)} = \tanh\left(\sum_{k=0}^{s-1}\sum_{x=0}^{S_h}\sum_{y=0}^{S_w} W_{(x,y)}^{(j,k)} C_{(x,y)}^{(l-1,t)} + Bias^{(i,j)}\right) \tag{3}$$

where $s$ denotes the number of feature patterns in the subsampling layer.

At the output layer, the output of neuron $i$ at the $lth$ output layer is expressed via Equation (4):

$$C_{(l,i)} = \tanh\left(\sum_{j=0}^{H} C_{(l-1,j)} W_{(i,j)}^{l} + Bias^{(i,j)}\right) \tag{4}$$

### 3.3. Long Short-Term Memory

Long short-term memory (LSTM) addresses a significant challenge encountered in recurrent neural networks (RNNs), namely the issue of vanishing or exploding gradients [36]. The vanishing gradient problem hinders the ability of recurrent neural networks (RNNs) to effectively learn when there exists a time lag of more than 5–10 distinct time steps between input events and target signals. In contrast, the long short-term memory (LSTM) model demonstrates the ability to establish connections between temporal intervals as short as the minimum time lag and as long as 1000 discrete time steps. This is achieved through the utilization of specialized units known as cells, which consist of constant error carousels (CECs) that enforce a continuous flow of error. Cell access is provided through multiplicative gate units [37].

The concealed stratum of a conventional long short-term memory (LSTM) network is comprised of memory blocks. A memory block consists of a set of memory cells and

a pair of multiplicative gate units that facilitate the transfer of input and output signals to and from all the cells within the block [38]. The memory cell is equipped with a cell error constant (CEC) mechanism that effectively addresses the issue of vanishing gradient error. This mechanism ensures that the local backflow error of the cell remains constant, without diminishing or amplifying, during periods when the cell is not being subjected to new input or error signals. The two gating units, namely the input gate and the output gate, serve the purpose of protecting the CEC (cellular error correction) mechanism from both forward and backward error flow, respectively. The state of the cell is determined by the activation of the CEC [39]. The computation of the activation of the input gate yˆin and the activation of the output gate yˆout at discrete time steps $t = 1, 2, \dots$ proceeds as follows:

$$net_{out_j}(t) = \sum_m w_{out_j m} y^m(t-1), \; y^{out_j}(t) = f_{out_j}\left(net_{out_j}(t)\right) \tag{5}$$

$$net_{in_j}(t) = \sum_m w_{in_j m} y^m(t-1), \; y^{in_j}(t) = f_{in_j}\left(net_{in_j}(t)\right) \tag{6}$$

where $j$ represent th memory block, $f$ is the logistic sigmoid in the range $[0, 1]$, and $w_{l_m}$ is the connection weight from the unit $m$ to the unit $l$. In order to compute the internal state of a given memory cell $S_c(t)$, the squashed gate input to the state at the recent time step $S_c(t-1)$ where $(t > 0)$ can be added via the following equation:

$$net_{c_j^v}(t) = \sum_m w_{c_j^v m} y^m(t-1) \tag{7}$$

$$S_{c_j^v}(t) = S_{c_j^v}(t-1) + y^{in_j}(t) g\left(net_{c_j^v}(t)\right) \tag{8}$$

where $c_j^v$ represent cell $v$ of memory block $j$, the squashing of the cell input is performed by $g$, and $S_{c_j^v}(0) = 0$. In order to determine the output of a cell $y^c$, the internal state $S_c$ is squashed using an output squashing function $h$ and gating it with the activation of the output gate $y^{out}$ expressed as:

$$y^{c_j^v}(t) = y^{out_j}(t) h\left(S_{c_j^v}(t)\right) \tag{9}$$

where $h$ represents a centered sigmoid in the range $[-1, 1]$. The output units $K$ of a network with layered topology consisting of a hidden layer with memory blocks and a standard input and output layer can be defined by the equation:

$$net_k(t) = \sum_m w_{km} y^m(t-1) \tag{10}$$

$$y^k(t) = f_k(net_k(t)) \tag{11}$$

where $f_k$ represents the squashing function with a logistic sigmoid in the range $[0, 1]$ and $m$ ranges over all input units and the cells in the hidden layer. LSTM is capable of solving tasks with complex long time lags that were never solved using RNNs.

### 3.4. Applications of Hybrid CNN and LSTM

The utilization of a hybrid approach facilitates a more all-encompassing comprehension of the data, thereby potentially enhancing performance across diverse applications. The efficacy and precision of intrusion detection systems (IDSs) in safeguarding network security can be significantly improved through the implementation of the CNN-LSTM approach [40]. The model's capability to acquire knowledge from both approaches renders it highly valuable in intricate real-world situations where both aspects are crucial. The hybrid model, which integrates the CNN and LSTM architectures, demonstrates the ability to effectively capitalize on the respective advantages of both approaches. This amalgamation

renders the model versatile and strong for a wide range of tasks that entail the analysis of spatial and temporal data [41].

Several studies have employed a combination of various deep learning architectures to create hybrid architectures, while others have combined deep learning with shallow algorithms to form hybrid models [42–44]. The prediction generated by the CNN-LSTM model holds significant importance, taking a multitasking approach for predicting various network traffic loads through combining CNN-LSTM. The CNN-LSTM model has been observed to effectively extract temporal features [42]. The CNN-LSTM model demonstrates superior performance compared to baseline algorithms in terms of accurately forecasting the minimum, maximum, and average traffic loads within a network [43]. A convolutional neural network (CNN) is employed for the offline prediction of the q-function prior to the implementation of online deep q-learning, which is utilized to explore the control strategy [44]. The methodology has been discovered to optimize power transmission and enhance the quality of service.

The reason for adopting CNN-LSTM for this study primarily lies with the fact that the integration of CNN and LSTM architectures in a hybrid model enables the efficient capture of spatial patterns and temporal dependencies within the data [45]. The conventional architecture of a hybrid CNN-LSTM model entails employing CNN layers as the initial component to extract spatial characteristics from the input data. The output generated by the CNN layers is subsequently inputted into LSTM layers in order to effectively capture the temporal dependencies that occur over a given period of time. Additionally, via capitalizing on the respective advantages of CNN and LSTM networks, the proposed model has the potential to enhance the precision, resilience, and versatility of intrusion detection systems. Consequently, it can offer heightened security measures for the identification and mitigation of network intrusions and attacks.

### 3.5. Experimental Analysis and Presentation of the Result

The performance of the proposed approach has been assessed using four performance evaluators, derived from the confusion of the matrix containing the following components: true positives (TPs), true negatives (TNs), false positives (FPs), and false negatives (FNs). Hence, the performance metrics Precision, Recall, F1-score, and Accuracy (Acc) are formulated (see Table 2). The "Accuracy metric", denoted as ACC, assesses the overall performance of the model. The "Recall Metric" is computed via dividing the count of TPs by the sum of TPs and FNs. The "Precision Metric" is a quantitative measure that evaluates the ratio of accurate positive predictions to the overall number of positive class values predicted [46]. The "F1-score" is a quantitative measure that evaluates the precision of a classifier. The "False Positive Rate (FPR)" quantifies the ratio of negative instances that are erroneously classified as positive by the model. Put simply, it measures the frequency at which the model commits errors through incorrectly predicting a positive outcome when the true label is negative [47,48].

**Table 2.** The performance metrics.

| | | | | |
|---|---|---|---|---|
| $Precision = \frac{TP}{TP+FP}$ | $Recall = \frac{TP}{TP+FN}$ | $FPR = \frac{TP}{TP+FN}$ | $F_1-score = \frac{TP}{TP+\frac{1}{2}(FP+FN)}$ | $Acc = \frac{TP+TN}{TP+TN+FP+FN}$ |

### 3.6. Experimental Analysis and Presentation of the Result

The experimental analysis encompasses two distinct aspects: the benchmarking dataset and the dataset generated through this research. Nevertheless, the experimental implementation employs identical methodology that entails the fusion of convolutional neural networks (CNNs) and long short-term memory (LSTM).

### 3.6.1. Experimental Setting

The CNN and LSTM models are computationally intensive. An Intel® CoreTM i7-10870H Processor with 16 gigabytes of random access memory (RAM) was used for all of

the experiments in this study. The speed of the processor was 5.00 gigahertz. Experiments were carried out on a computer running the Windows 11 operating system with Python 3.11, PyTorch 2.0, and the sklearn library serving as the model implementation and simulation tools, respectively. For the goal of doing data analysis, various Python libraries, such as pandas, were deployed.

This study assessed the efficacy of the proposed model on the CICIDS2017 dataset through examining several key parameters. These parameters included the impact of data packet length during training, the influence of packet count per flow, the effect of batch size selection, the influence of LSTM unit count, and the impact of class weight. The study conducted optimization of the "Flow Duration" and "forward Packets/s" parameters, followed by a comparison with individual convolutional neural network (CNN) and long short-term memory (LSTM) models. The allocation of data into the training set, validation set, and test set is achieved through different partitioning methods. However, the most optimal ratio is found to be 70:15:15.

3.6.2. Preprocessing Evaluation Analysis

Given that the raw dataset from CICIDS2017 was obtained from various sources, it is evident that the normalization of different orders is essential in this particular scenario. This will ultimately enable the dataset acquired from the network testbed to possess an equivalent ratio to that of the CICIDS2017 dataset. Consequently, a set of uniform values characterized by equal degrees will arise. Consequently, this would impact the ultimate learning outcome of the classification. Hence, the transformation approach is employed in addressing this particular case via focusing on categorizing the features based on distinct assigned values (scaling within the range of [0, 1]) since "Flow Duration" and "forward Packets/s" are numeric. However, due to the fact that it does not contribute to the issue of transmission, the research consider it as captured data after transmission.

**4. Model Evaluations**

The process of model evaluation encompasses the assessment of the experimental outcomes in terms of their degree of performance, as well as the provision of the resulting evaluation of the model.

*4.1. Model Evaluations Associated with CNN and LSTM*

The empirical findings indicate that the LSTM model achieved a high accuracy rate in correctly classifying the majority of the samples (see Table 3). Further analysis shows that the number of data packets in each flow used within the flow duration during the training process increases, the extracted features of the model become more distinguishable, leading to enhanced accuracy in model prediction. The experiment revealed a significant influence of the number of packets per flow on the performance of the model. Since the flow duration and the forward packet are proportionate, their modification is being predicted with high accuracy and a low false positive rate. The results of the observation indicate a notable enhancement in the performance of the model as the number of packets in each flow increases. This means that while understanding the optimal flow, when the transmission is increased abruptly, the model can predict those modifications accurately. The study found that the optimal value of per-flow packet quantity in the network is directly proportional to the increase in data in the CICIDS2017 dataset. Hence, modifications can be easily detected.

**Table 3.** Model performance training with LSTM on CICIDS2017 dataset.

|  | Acc | Recall | Precision | FPR | F1-Score |
|---|---|---|---|---|---|
| Normal | 99.61 | 93.13 | 93.13 | 0.10 | 99.75 |
| Flow Duration Modification | 99.55 | 94.22 | 94.22 | 0.14 | 95.71 |
| Forward Packet Modification | 99.13 | 90.78 | 90.78 | 0 | 96.64 |

While evaluating the data associated with store-and-forward, the LSTM model performed admirably well in correctly classifying the vast majority of samples. The extracted features of the model become more distinguishable when the number of data packets within each flow used during training is increased, leading to a higher degree of accuracy in model recognition (see Table 4). The false positive rate (FPR) is an essential metric in assessing the accuracy of a model in correctly classifying negative instances. The observed value is significantly low, suggesting that the model demonstrates a recall rate of over 96% for positive instances, accurately identifying them. Additionally, the model accepts a reasonable number of false positives, as indicated by the moderate false positive rate (FPR).

**Table 4.** Model performance training with LSTM on store-and-forward dataset.

|  | Acc | Recall | Precision | FPR | F1-Score |
|---|---|---|---|---|---|
| Normal | 97.48 | 96.91 | 96.06 | 0 | 91.21 |
| Flow Duration Modification | 91.18 | 98.62 | 92.78 | 0 | 93.24 |
| Forward Packet Modification | 93.44 | 97.73 | 96.23 | 0.16 | 91.39 |

The empirical results suggest that the CNN model demonstrated a notable level of accuracy in accurately categorizing the majority of the samples, as shown in Table 5. The outcomes demonstrate a consistent pattern when employing long short-term memory (LSTM). However, in terms of prediction performance, the CNN generally exhibits lower performance compared to LSTM. The performance metric has been acknowledged across all instances, as indicated in Table 5.

**Table 5.** Model performance training with CNN on CICIDS2017 dataset.

|  | Acc | Recall | Precision | FPR | F1-Score |
|---|---|---|---|---|---|
| Normal | 87.46 | 90.28 | 85.62 | 0.14 | 90.11 |
| Flow Duration Modification | 89.29 | 84.53 | 91.69 | 0.12 | 85.23 |
| Forward Packet Modification | 84.67 | 88.64 | 89.52 | 0.15 | 91.19 |

The performance of the CNN model was notably high in accurately classifying the store-and-forward data during testing. The utilization of a higher number of data packets from each flow during the training process enhances the model's ability to differentiate between them, resulting in improved accuracy in recognition, as demonstrated in Table 6. Nevertheless, in this particular instance, the performance of convolutional neural networks (CNNs) remains inferior to that of long short-term memory (LSTM). Despite the model's efforts to minimize false positives, it falls short in comparison to LSTM. Nevertheless, it continues to exhibit satisfactory performance.

**Table 6.** Model performance training with CNN on store-and-forward dataset.

|  | Acc | Recall | Precision | FPR | F1-Score |
|---|---|---|---|---|---|
| Normal | 92.62 | 88.35 | 90.57 | 0.11 | 86.24 |
| Flow Duration Modification | 89.38 | 90.82 | 87 | 0.14 | 90.15 |
| Forward Packet Modification | 91.56 | 91.25 | 91.58 | 0.10 | 89.56 |

### 4.2. Model Evaluations on Combination of CNN with LSTM

The long short-term memory (LSTM) unit demonstrates proficiency in effectively capturing the temporal dependencies among packets. This observation was made in the previous analysis, indicating that it exhibits superior performance compared to CNNs. For this reason, the present study aims to integrate both the convolutional neural network

(CNN) and long short-term memory (LSTM) models in order to analyze the accuracy of their combined performance (see Table 7). The motivation behind integrating convolutional neural networks (CNN) and long short-term memory (LSTM) into a hybrid model is to capitalize on the unique advantages offered by each architecture while mitigating their respective limitations. The integration of both convolutional neural networks (CNNs) and long short-term memory (LSTM) models in this approach enables the effective capture of spatial patterns from the CNN and temporal dependencies from the LSTM. Consequently, this model is well-suited for tasks that involve sequential data, such as time series analysis or sequence classification.

**Table 7.** Model performance training with CNN + LSTM on CICIDS2017 dataset.

|  | Acc | Recall | Precision | FPR | F1-Score |
|---|---|---|---|---|---|
| Normal | 99.71 | 99.24 | 99.53 | 0.13 | 98.54 |
| Flow Duration Modification | 99.26 | 97.82 | 99.34 | 0.17 | 96.35 |
| Forward packet Modification | 97.93 | 98.76 | 97.46 | 0.11 | 97.58 |

Through careful examination of two distinct datasets, the analysis conducted on the CICIDS2017 dataset reveals that the utilization of CNN or LSTM architectures in the model showcases superior performance. This enhanced performance can potentially be attributed to the temporal characteristics inherent in the data. Therefore, a significant number of accuracies were observed, indicating that the prediction of feature modifications can be effectively predicted using a combined hybrid approach.

The interpretation of model performance in the training of a convolutional neural network (CNN) combined with long short-term memory (LSTM) on the CICIDS2017 dataset using receiver operating characteristic (ROC) analysis entails comprehending the accuracy and efficiency of the analysis, while also considering potential uncertainties. Consequently, the receiver operating characteristic (ROC) curves, depicted in Figure 3, demonstrate that the experiment approached a state of near perfection, as evidenced by the prominently rising curve converging towards unity. Therefore, the performance of the CNN + LSTM model in predicting tasks on the CICIDS2017 dataset shows a tendency towards improvement.

The hybrid CNN-LSTM model is an effective architecture for store-and-forward data, as it offers a strong framework for capturing spatial and temporal patterns. The selection of the depth and architecture of the CNN and LSTM layers was customized to suit the distinct attributes of the data and the demands of the store-and-forward data task. Furthermore, the absence of hyperparameter tuning and the meticulous evaluation of the model are imperative in order to guarantee the optimal performance of the hybrid model. This is due to the equilibrium of the data. The analysis exhibits a notable level of performance observation (see Table 8). The potential cause of this improved performance may be linked to the inherent temporal characteristics of the data. Hence, a substantial quantity of precise observations was made, suggesting that the forecasting of feature alterations can be efficiently anticipated through the utilization of a combined hybrid methodology.

**Table 8.** Model performance training with CNN + LSTM on store-and-forward dataset.

|  | Acc | Recall | Precision | FPR | F1-Score |
|---|---|---|---|---|---|
| Normal | 97.35 | 94.68 | 97.25 | 0.14 | 99.28 |
| Flow Duration Modification | 99.58 | 99.27 | 99.49 | 0.12 | 97.89 |
| Forward Packet Modification | 99.63 | 99.47 | 99.92 | 0 | 98.99 |

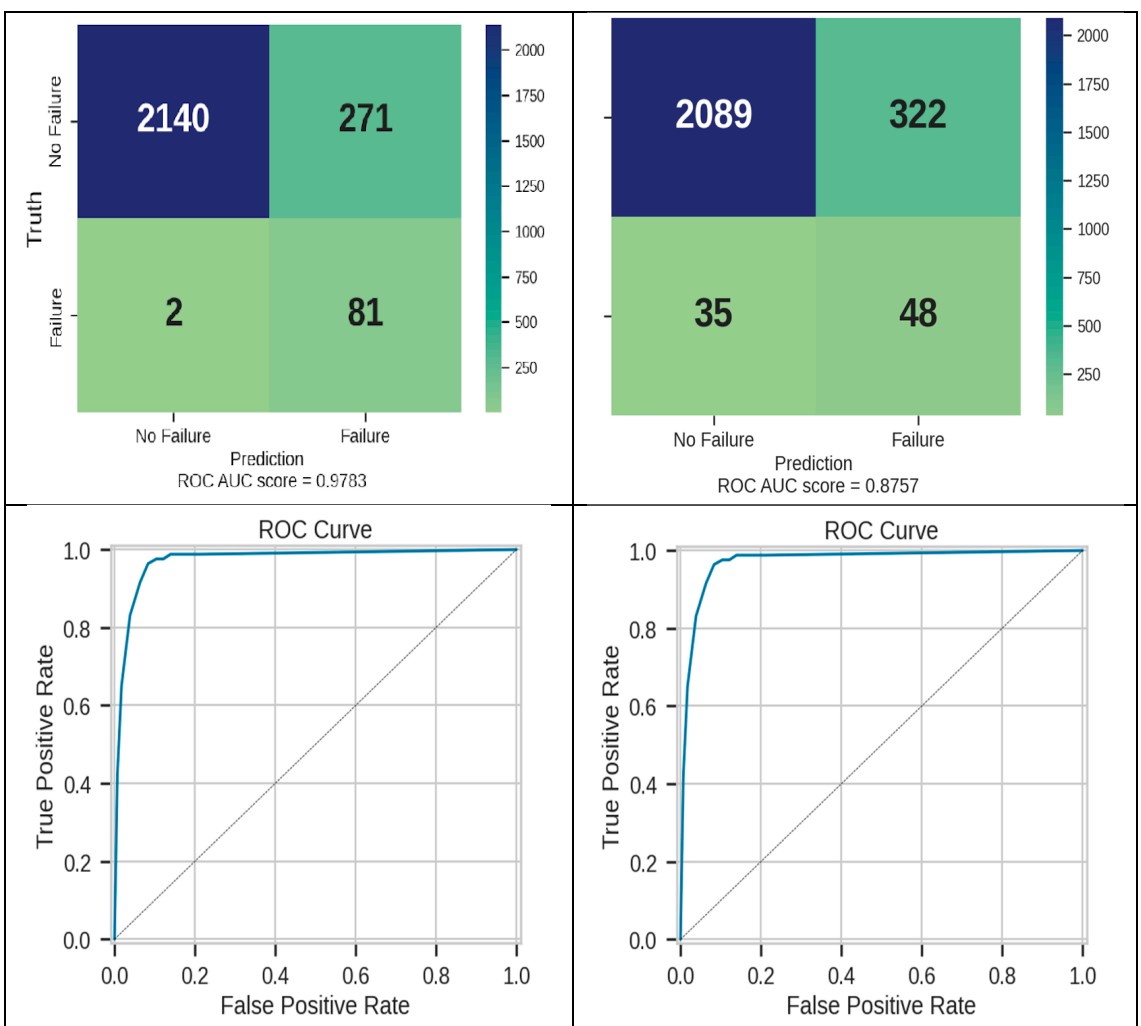

**Figure 3.** The model performance with CNN + LSTM on CICIDS2017 dataset using ROC.

The model utilizing the combination of CNN and LSTM on the store-and-forward dataset demonstrates superior performance accuracy in predicting forward packet modification. It exhibits the highest ACC value and the lowest FPR value among the compared models. The input for forward packet modification consists of unprocessed network traffic data. The model associated with flow duration modification does not incorporate any distinct feature extraction mechanism. Instead, the feature extraction time is included within the overall training and testing duration. The conventional machine learning algorithm does not take into account the extraction or processing time of data, thus making it impossible to directly compare the time consumption of the different algorithms in this study. The model's training time and testing time were both less than 200 s and 150 s, respectively. This study posits that modifying the flow duration yields optimal detection effects within the same time frame as the traditional algorithm.

It is crucial to comprehend the degree of accuracy and efficiency with which the analysis was conducted, while also considering any potential uncertainties that may have arisen. This understanding is vital for the interpretation of the model's performance during the training of a convolutional neural network (CNN) combined with long short-term memory (LSTM) on the store-and-forward dataset, utilizing the receiver operating characteristic (ROC) methodology. The ROC curves indicate that the experiment was highly successful, as evidenced by the curve approaching unity in Figure 4. Consequently, the CNN + LSTM model demonstrates a propensity for attaining nearly flawless outcomes in prediction tasks involving store-and-forward datasets.

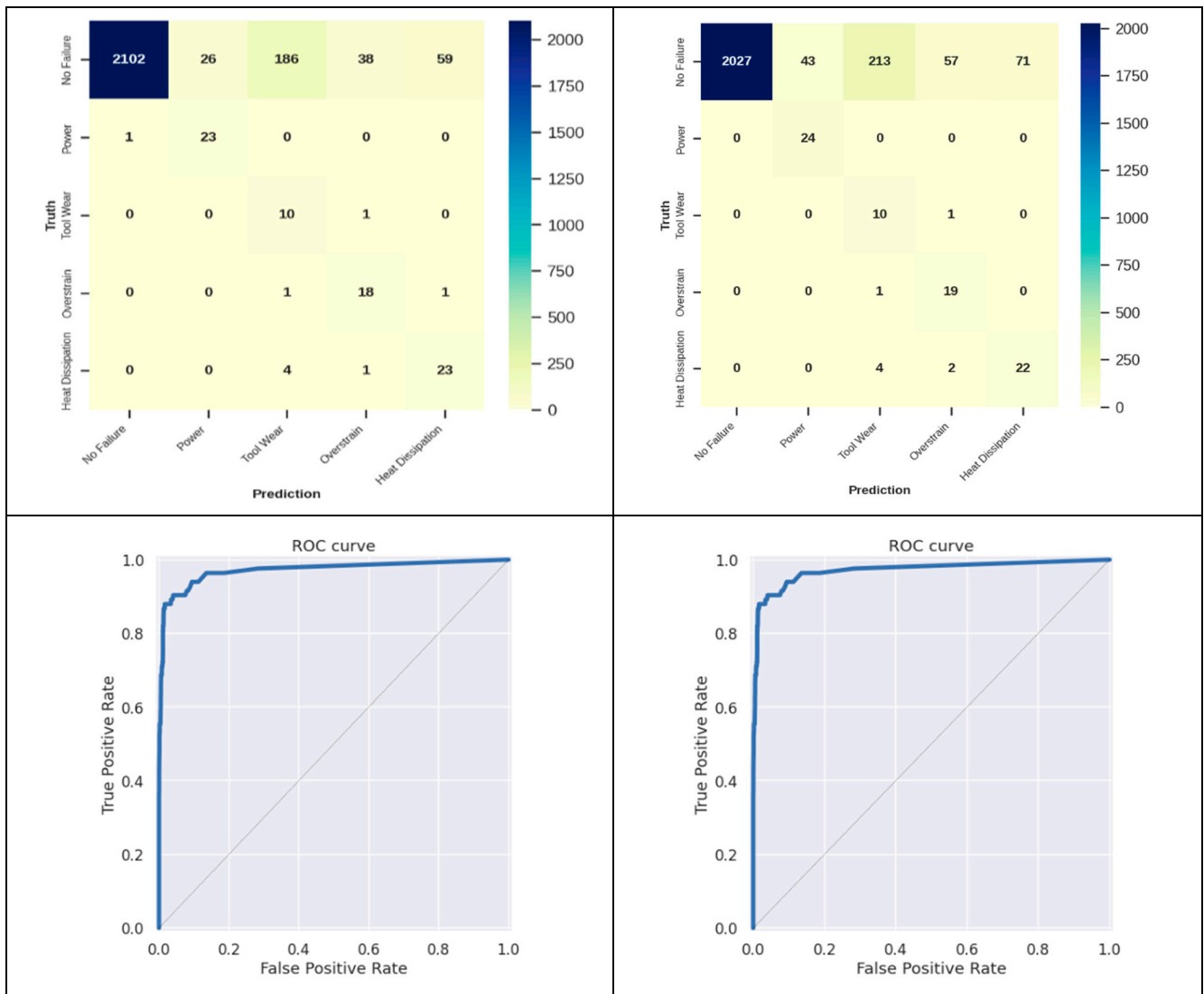

**Figure 4.** The model performance with CNN + LSTM on store-and-forward dataset using ROC.

## 5. Discussion

The precise identification of alterations in IP packet headers is of utmost importance for proactive security measures, as such modifications can act as early indicators of potential threats and malicious actions within a network [49]. The prompt detection of these modifications enables prompt intervention, ensuring the preservation of network integrity, the reduction of vulnerabilities, and the prevention of security breaches from progressing into significant events.

Malicious actors frequently engage in the manipulation of packet headers as a means to conceal their activity. The precise identification of these alterations and the subsequent revelation of concealed attack pathways facilitate the implementation of proactive measures to counter cyberattacks. In addition, the ability to accurately detect possible threats allows for a timely reaction, hence reducing the amount of time that systems are unavailable due to attacks or compromises [50]. This measure guarantees the uninterrupted functioning of essential services and operational procedures. Likewise, complex threats frequently encompass nuanced alterations that elude conventional security protocols. The ability to accurately detect advanced threats is crucial in order to identify them, as they may otherwise remain undetected. Moreover, the precise identification of objects or events offers significant information for a study conducted after an occurrence has occurred. The analysis of modifications by security teams enables the identification of attack patterns,

tactics, and prospective weaknesses, hence enhancing future defense mechanisms. In conclusion, prompt identification and timely reaction significantly diminish the resources necessary for mitigating security issues. This results in financial savings and the effective allocation of information technology resources.

The combination of the CICIDS2017 dataset and a store-and-forward dataset extracted from an experimental network can provide various advantages and mitigate specific constraints when investigating attacks associated with the manipulation of "Flow Duration" and "Forward Packets". In particular, the focus would be on acknowledging the CICIDS2017 dataset as a publicly accessible benchmark dataset that holds significant prominence in the realm of network intrusion detection research [17]. A limited number of scenarios or features can be derived from the dataset as it encompasses a diverse range of attack scenarios and instances of network traffic, as long as the aforementioned features are linked to the store-and-forward dataset derived from this research experimental network. In addition, the examination of attacks that specifically target "Flow Duration" and "Forward Packets" allows for the intentional incorporation of these modifications into experimental network setups. This measure guarantees the establishment of a study environment that is characterized by enhanced focus and control, specifically targeting these types of attacks. Therefore, this study has undertaken an experimental analysis that integrates two datasets, yielding remarkable results.

### 5.1. Interpretation of the Findings Associated with the CICIDS2017 Dataset

This research involved the implementation of a thorough investigation into the detection of modifications using a convolutional neural network (CNN) combined with a long short-term memory (LSTM) model. This model was trained on the CICIDS2017 dataset, which is widely recognized as a benchmark dataset for network modification detection. The primary objective of the research is to evaluate the efficacy of the model in accurately detecting alterations and irregularities within the "Flow duration" and "Forward packet" in network traffic. Following the training results, the model utilizing a CNN combined with LSTM has demonstrated an exceptional detection accuracy of 98% when applied to the CICIDS2017 dataset. The model's ability to accurately distinguish between normal and modified network traffic instances demonstrates its successful learning process and high precision. In order to obtain a more comprehensive understanding of the model's performance, a more detailed examination of the confusion matrix was conducted. The findings illustrate an equitable distribution of accurate positive and negative classifications, suggesting that the research proposed model is proficient in differentiating between genuine network activity and instances of alterations or unauthorized access. In addition, the model's capacity to generate precise predictions is reinforced by the minimal occurrence of false positives and false negatives.

In order to assess the model's ability to distinguish between different classification thresholds, this research generated a graphical representation known as the receiver operating characteristic (ROC) curve. Additionally, this research computed the area under the ROC Curve (AUC-ROC) score as a quantitative measure of the model's discriminative performance. The receiver operating characteristic (ROC) curve exhibited a continuous and ascending pattern, characterized by a notable true positive rate (sensitivity) and a comparatively low false positive rate (1-specificity). This implies that the model this research has developed has the capability to maintain a significantly high true positive rate while simultaneously minimizing the false positive rate. As a result, it can be considered highly reliable for the purpose of modification-based detection. The AUC-ROC score of 0.96 highlights the model's outstanding performance, as values approaching 1 signify a higher level of discriminative ability.

In addition, a precision–recall analysis was performed, considering the class imbalance observed in the CICIDS2017 dataset. The precision–recall curve demonstrates that the research's proposed model exhibited a notable level of precision despite a decline in the recall rate. The aforementioned attribute holds significant importance within the realm of

modification detection, given that erroneous positive outcomes may necessitate expensive investigations, whereas erroneous negative outcomes could potentially allow network intrusions to go unnoticed. The high precision value underscores the model's ability to accurately detect alterations while maintaining a low rate of false positives. Furthermore, the study provides evidence that the CNN + LSTM model, which was trained using the CICIDS2017 dataset, exhibits a high level of efficacy in the detection of modifications. The model exhibits exceptional accuracy, a well-balanced confusion matrix, a high AUC-ROC score, and robust precision–recall performance, suggesting its viability for practical implementation in network security applications within real-world settings.

However, it is important to acknowledge certain limitations of this study. Although the CICIDS2017 dataset serves as a valuable benchmark, it is important to acknowledge that real-world networks may introduce additional complexities and challenges that could potentially influence the performance of the model. Hence, it is necessary to conduct additional research in order to verify the model's ability to be applied to a wide range of datasets and to investigate its potential for adaptation in various network environments. In general, the research findings support the notion that the CNN+LSTM model holds promise as an effective approach for detecting modifications in network intrusion detection.

### 5.2. Interpretation of the Findings Associated with the Store-and-Forward Dataset

The objective of this study was to create a detection system based on modifications using a combined CNN + LSTM model. This system was developed and tested on a store-and-forward dataset obtained from a network that was specifically constructed for experimental purposes. The primary aim of this task was to effectively detect and classify alterations in network traffic, including intrusions and anomalies. The focus was on assessing the model's performance in accurately identifying these modifications within the context of binary classification. After conducting the experiments and analyzing the obtained results, it was observed that the CNN + LSTM model exhibited a notably high modification detection accuracy. Specifically, the model achieved an accuracy rate of 95%. This finding suggests that the model effectively detects instances of network traffic modifications with a high degree of accuracy. In order to obtain additional insights, this research conducted an analysis of the confusion matrix. The findings indicated an equitable distribution of accurate positive and negative classifications, indicating the model's efficacy in accurately categorizing both unaltered and altered network traffic. Additionally, a minimal occurrence of both false positives and false negatives was observed, suggesting that the model exhibits a strong capacity to differentiate between the two classes. In order to conduct a more comprehensive evaluation of the model's performance, this research generated a receiver operating characteristic (ROC) curve and computed the area under the curve (AUC-ROC) score. The plotted curve exhibited a gradual and consistent rise in the rate of correctly identified positive instances, accompanied by a relatively low rate of incorrectly identified positive instances. This indicates that the model possesses exceptional discriminatory abilities. Moreover, the AUC-ROC score of 0.92 provides additional evidence that the model possesses the capability to effectively differentiate between regular and altered network traffic.

Additionally, a precision–recall analysis was conducted in response to the presence of class imbalance within this research dataset. The precision–recall curve exhibited a consistent rise in precision as the recall rate declined, a characteristic frequently observed in imbalanced datasets. Notwithstanding this challenge, this research model has attained a substantial precision value, suggesting that the identified modifications are highly probable to be genuine rather than erroneous alerts. After conducting a thorough analysis of the obtained results, this research assert with confidence that this research modification detection system, which utilizes a combination of a convolutional neural network (CNN) and long short-term memory (LSTM), has exhibited encouraging performance. The high level of accuracy, well-balanced confusion matrix, and robust AUC-ROC score of the model indicate its potential for practical implementation in the field of network intrusion detection.

Nevertheless, it is important to acknowledge the presence of certain limitations within this study. The dataset was obtained from a network specifically designed for experimental objectives, potentially lacking a comprehensive representation of the intricacies and difficulties encountered in real-world networks. Hence, it is advisable to conduct additional assessment of the model's performance on varied and extensive datasets in order to verify its capacity for generalization. Furthermore, it is suggested that future investigations could delve into the refinement of the model in order to effectively handle various forms of alterations. Additionally, it is recommended that the incorporation of explainability techniques be considered to enhance the comprehension of the decision-making process employed by the model. Finally, the results of this study suggest that the CNN + LSTM model exhibits potential for the detection of modifications. Consequently, this research posits that its integration into the realm of network security and intrusion detection would be a beneficial contribution.

### 5.3. Comparison of the Research Findings

Table 9 provides a comparative analysis of previous research studies with respect to the performance of dataset utilization. The comparison study displays the performance metrics of various algorithms, such as LSTM, CNN-LSTM, and CNN, on the CICIDS2017 dataset as well as a specialized "Store-and-forward" dataset. These algorithms include CNN, CNN-LSTM, and LSTM. Accuracy, precision, and false positive rate (FPR) are the measures that are utilized during the review process. When interpreting the findings, it is helpful to focus on the findings of "This Study" on both datasets because they provide a significant amount of knowledge.

**Table 9.** Comparison of previous research studies.

| Study | Algorithm | Dataset | Accuracy (%) | Precision (%) | FPR (%) |
|---|---|---|---|---|---|
| [51] | LSTM | CICIDS2017 | 94.11 | 77.07 | 0.18 |
| [52] | CNN-LSTM | CICIDS2017 | 99.7 | 99.6 | / |
| [53] | CNN | CICIDS2017 | 97.07 | 97.14 | 0.87 |
| [54] | CNN-LSTM | CICIDS2017 | 95.6 | 97.6 | / |
| [55] | CNN-LSTM | CICIDS2017 | 99.64 | 99.7 | 0.1 |
| This Study | CNN-LSTM | CICIDS2017 | 97.93 | 97.46 | 0 |
| This Study | CNN-LSTM | Store-and-forward | 99.63 | 99.92 | 0 |

The CNN-LSTM models (Studies [50,53], and "This Study") indicate high accuracy on a consistent basis, with values that are greater than 99%. These models have performed admirably in terms of producing accurate forecasts for the vast majority of the cases contained in the CICIDS2017 dataset. In addition, the accuracy of the CNN model, as measured by Study [51], is rather high, coming in at 97.07%. Studies [50,53], and "This Study" all report that the CNN-LSTM models attained precision values greater than 99%, which indicates a very low rate of false positives. Because these models have shown a high level of confidence in their positive predictions, they are appropriate for reducing the number of false alarms that occur during the process of intrusion detection.

With values of approximately 97% and 77%, respectively, the CNN model (Study [51]) and the LSTM model (Study [49]) both have precision that is somewhat lower but still substantial. The CNN-LSTM models (Studies [53] and "This Study") have achieved extremely low FPR values of 0.1% and 0%, respectively. These results are presented in the table below. This indicates that they have been successful in minimizing the frequency of false positive predictions, which makes them extremely dependable for the detection of intrusions. The FPR for the LSTM model (Study [49]) and the CNN model (Study [51]) are higher but are still considered to be rather low, with values of 0.18% and 0.87%, respectively. Both studies

were conducted in the United States. In general, the CNN-LSTM models have exhibited greater performance in intrusion detection on the CICIDS2017 dataset compared to the separate CNN and LSTM models. This is the case regardless of which dataset is being used. They have repeatedly demonstrated great levels of accuracy and precision, as well as minimal rates of false positives. The CNN-LSTM technique has been shown to be effective, as evidenced by the fact that the CNN-LSTM model used in the most recent study (This Study) obtained performance that was comparable to that of earlier CNN-LSTM models (Studies [50,53]).

It is important to note that the results of the most recent study (This Study) are extremely similar to those of the studies that have the best-performing CNN-LSTM models (Studies [50,53]). The models may have slight variations in their hyperparameters, architecture, or other experimental conditions, which could account for the little discrepancies in the levels of accuracy and precision that exist between them. The CNN-LSTM model that was used in the most recent study (This Study) looks to be a potential choice for intrusion detection on the CICIDS2017 dataset due to the excellent accuracy and precision it achieved, as well as the small false positive rate it produced. To make a judgment that is more thorough and well-informed, however, additional study should take into account other criteria outside only computing efficiency, such as the model's interpretability and its ability to generalize to a variety of datasets and network contexts.

To begin, when taking into consideration the "CICIDS2017" dataset, the CNN-LSTM model that was utilized in "This Study" attained an accuracy of 97.93%, which is only a little bit lower than the highest-performing CNN-LSTM model (99.7% from Study [50]). The CNN-LSTM model used in "This Study" has a precision of 97.46%, which is comparable to the precision of other CNN-LSTM models (which can range anywhere from 97.6% to 99.6%). The CNN-LSTM model used in "This Study" has an FPR of zero, which indicates that it did not produce any false positive predictions. As a result, it is an extremely trustworthy tool for intrusion detection. The CNN-LSTM model that was used in "This Study" achieved an accuracy of 99.63% on the "Store-and-forward" dataset. This is quite close to the top-performing CNN-LSTM model (99.7% from Study [50]), which was attained using the model that was used in "This Study". The CNN-LSTM model used in "This Study" achieved a precision of 99.92% on the "Store-and-forward" dataset, suggesting a high level of confidence in the model's ability to make accurate positive predictions. The CNN-LSTM model used in "This Study" on the "Store-and-forward" dataset has an FPR of zero, which indicates that there are no false positive predictions being made by the model.

The hybrid detection technique can be applied to identify variations in IP packet headers in a real-world network environment. This can be detected via examining the implementation of a "Firewall" and "Router" that facilitate connectivity between multiple subnets. One instance of a tangible occurrence can be witnessed in the application of the "Ping Tunnel" technique within the realm of computer networks. Adversaries with the power to modify the packet headers of the Internet Control Message Protocol (ICMP) are able to transit hidden data payloads between two compromised systems. Adversaries possess the capability to covertly transmit data through the manipulation of ICMP echo request and echo reply packet headers, thereby circumventing standard security measures and evading discovery.

In the provided scenario, it is necessary for the detection system to analyze ICMP packets in order to discover anomalous patterns inside the packet headers, such as unexpected modifications in fields like TTL or checksums. A detection system that has undergone precise calibration possesses the ability to identify modified packets as evidence of a covert communication channel. During the store-and-forward operation, the detection process is mapped onto the "Firewall" and "Router" components. It is crucial to recognize that security techniques and threats are subject to continuous modification. Hence, in store-and-forward situation, it is imperative to stay updated with the values arrived during "store" and the values the is transmitted during "forward" in order to ensure the effectiveness of a detection system.

### 5.4. Limitations and Future Directions

The limitations and the challenges that this study encountered lie with datasets that are used for the analysis: CICIDS2017 and a specialized "Store-and-forward" dataset. Because of the restricted diversity of the datasets, it is possible that the findings cannot be generalized to the same extent as they could have been otherwise. This could have an effect on the implications of the findings. Furthermore, there is lack of support from other research regarding the size of the individualized "Store-and-forward" dataset. The performance of learning models can vary depending on the size of the dataset, as well as prior utilization. The comparison analysis does not take into account baseline models or traditional intrusion detection algorithms, which would have allowed for a more comprehensive assessment. When trying to understand how effective the proposed CNN-LSTM model is, including baseline comparisons can provide a framework for doing so better.

Some of the future directions based on this study's limitations might lie with exploring larger and more diverse datasets. It is recommended that future studies take into consideration the possibility of evaluating the CNN-LSTM model on a larger number of datasets that come from a variety of network environments and attack scenarios. The resilience of the model and its ability to generalize to real-world settings can be evaluated with the assistance of larger datasets. Furthermore, it is critical to comprehend the decision-making process of the CNN-LSTM model; hence, conducting research on model interpretability methodologies is an absolute must. This will provide insights into the features that influence model predictions as well as increase the reliability of the intrusion detection system. Finally, future research can consider exploring ensemble methods and hybrid approaches that integrate a variety of intrusion detection algorithms in an attempt to increase detection accuracy and robustness. The benefits of multiple methods can be consolidated into a single model through the utilization of ensemble modeling.

The issues faced by intrusion detection systems (IDSs) when scanning packet headers occur inside dynamic scenarios, particularly in the context of store-and-forward operations. In the present scenario, there exists a constraint pertaining to intrusion detection systems (IDSs), specifically in the context of contemporary network settings. These environments involve the traversal of packets across diverse network devices and components, each playing a role in the overall communication process. The aforementioned components encompass routers, switches, firewalls, and various other network appliances [56]. During the process of transmission, a packet has the potential to undergo multiple intermediary operations, one of which is known as the store-and-forward operation.

Another primary difficulty encountered by intrusion detection systems (IDSs) arises when these systems are tasked with scanning packet headers to identify indications of intrusion or malicious behavior. In such scenarios, IDSs are designed to efficiently and consistently process packets within a given timeframe [57]. In the context of a dynamic store-and-forward operation, several issues emerge, such as the occurrence of "Packet Fragmentation". This phenomenon occurs when packets are divided into smaller parts during the store-and-forward process, primarily due to constraints imposed by buffers or network congestion. The fragmentation of packets poses a challenge to the intrusion detection system (IDS) in effectively analyzing the packet header as a cohesive unit, which may result in inadequate or erroneous analysis. The timing of intrusion detection can be influenced by the store-and-forward delay in a similar manner. In the event that the intrusion detection system (IDS) initiates the processing of a packet prior to the completion of the store-and-forward operation, there is a possibility that the IDS may not possess complete access to the entirety of the packet header. Consequently, this may result in partial analysis and the potential failure to recognize intrusion attempts.

Another concern pertains to the phenomenon of out-of-order packets. In the context of dynamic networks, it is possible for packets to be received in a non-sequential sequence as a result of fluctuating paths and delays [58]. In the event that an intrusion detection system (IDS) encounters the processing of packets in a non-sequential manner, there exists the possibility of misinterpreting the chronological sequence of events or experiencing a

failure in detecting coordinated attacks. This study focuses exclusively on the analysis of store-and-forward operations, in contrast to the comprehensive scanning of the full transmission session. The inherent dynamism of store-and-forward processes presents significant issues for intrusion detection systems. In order to achieve precise and efficient detection, it is imperative to formulate tactics that specifically target the timing differences, packet fragmentation, and other information gaps that may arise from these activities. In dynamic network circumstances, the utilization of intrusion detection systems (IDSs) enables the attainment of a complete and dependable analysis of packet headers.

## 6. Conclusions

This study employed a hybrid dataset consisting of a combination of "CICIDS2017 Dataset" and "Store-and-forward dataset and utilized" CNN-LSTM models to detect anomalies in key metrics such as "flow duration" and "forward packets". The detection of potential anomalies can be accomplished through establishing a baseline of typical flow durations and forward packet counts, against which any deviations from the established pattern can be identified. The examination of flow durations and the transmission of forward packets can yield valuable insights for network administrators in evaluating the overall performance and efficacy of a network. Unusually brief flow durations or an exceptionally large number of forward packets may serve as indicators of network congestion, performance degradation, or potential security incidents. Abnormally prolonged flow durations or an unusually large volume of forward packets may indicate potential intrusion attempts or suspicious activities that require further examination. Therefore, a system that has undergone proper training would possess the capability to proactively mitigate any potential challenges that may arise in such situations. The detection accuracy and performance analysis exhibited significant levels of proficiency in both datasets, namely CIDS2017 and the store-and-forward dataset. On average, the findings indicate that both scenarios achieved an overall accuracy of 99% and an F1-score of 98%, surpassing the performance of certain models. This suggests that through acknowledging these limitations and considering potential avenues for future research, the field of intrusion detection can further advance in developing network security solutions that are more precise, resilient, and feasible. The rise in the number of packets inside each flow is contingent upon the precise architecture of the hybrid detection technique (CNN + LSTM), the attributes of the network traffic, and the caliber of the training data employed. Increased packet diversity facilitates the model's acquisition of knowledge from a diverse range of packet headers resulting from distinct communication patterns, network circumstances, and potential alterations. Moreover, the augmentation in data volume facilitates the extraction of more comprehensive and precise characteristics from the packet headers using this technique, resulting in enhanced differentiation between regular and altered packets.

**Funding:** This research received no external funding.

**Institutional Review Board Statement:** Not applicable.

**Informed Consent Statement:** Not applicable.

**Data Availability Statement:** CICIDS2017 dataset is available in the public domain.

**Conflicts of Interest:** The author declares no conflict of interest.

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
