# Peer review of "Hybrid Detection Technique for IP Packet Header Modifications Associated with Store-and-Forward Operations"

_applsci, doi:10.3390/app131810229_

Round 1

Reviewer 1 Report

The paper addresses a critical issue in network security - the detection of IP packet header modifications within a network that employs a store-and-forward operation. The author has proposed a novel approach that combines Convolutional Neural Networks (CNN) and Long Short Term Memory (LSTM) to predict and detect modifications to packet headers that might otherwise go unnoticed by traditional intrusion detection systems (IDS).

The author provides a clear overview of the problem being addressed. It emphasizes the limitations of existing research, which primarily focuses on IDS inspecting the entire network transmission session but overlooks the potential for modifications to packet headers during the store-and-forward process. The approach of employing a hybrid model of CNN and LSTM is intriguing and well-motivated, particularly with the observed performance improvement as the number of packets within each flow increases, leading to an impressive average detection performance of 99%.

In my personal opinion, this work significantly contributes to the field of network security, specifically in identifying IP packet header modifications in the context of the store-and-forward operation. The combination of deep learning techniques, as demonstrated by the CNN-LSTM hybrid, shows promise in handling cases where transmission patterns change abruptly.

However, below are some issues that need to be addressed before proceed with the publication of the paper:

  1. Consider providing a concise statement about the importance of accurately detecting modifications in IP packet headers, especially in the context of security and potential threats.

  2. The author needs to explain how the proposed methodology addresses the limitations of existing research that primarily relies on intrusion detection systems (IDS)?

  3. The author needs to elaborate on the factors contributing to the observed 99% average detection performance when the number of packets within each flow increases?

  4. You need to provide details on the practical implementation of this detection technique in a real-world network setting?

Here are some typos and grammar comments:

This implies that when comprehending the ideal pattern, the model exhibits accurate 21 predictions for medications in cases where the transmission is abruptly increased.

I think you mean modification, not medication? Am I right?

The order of the discussed related works is incorrect. The discussion has to be in chronological order in which the oldest work should be presented first.

A summary of the related works identifying the research gap and the importance of the issue should be given at the end of the related work section.

The dataset for this study are generated from two different approach

The dataset for this study <is>, not are.

To facilitate the analysis of the packet during the transmission session and generate the dataset

The above sentence seems to be incomplete. You should add “…. and generate the dataset the following steps were performed.

The equations listed in subsection 3.2. Convolutional Neural Network need some supporting references. Also, it is not clear which version of CNN the author is referring to in this section. The CNN algorithm was a continuous development and the version that you are referring to should be clearly stated.

Likewise, the equations in subsection (3.2. Long Short Term Memory) have to be supported by the relevant references.

Section 3.2 should be numbered 3.3

3.3. Applications of Hybrid CNN an LSTM

The above section should be numbered 3.4 and d is missing from an

3.4. Performance Metrics of The Hybrid deep learning architecture

Should be 3.5

3. 5. Experimental Analysis and Presentation of the Result

Should be 3.6. The numbering of its subsections should be corrected as well.

There are many typos and grammar mistakes. I would recommend to double-check the language of the paper. 

Author Response

Reviewer 1

Issues 1

  1. Consider providing a concise statement about the importance of accurately detecting modifications in IP packet headers, especially in the context of security and potential threats.

Response to the Issues 1

Thank you very much for raising this issue and I responded to the comment below

The precise identification of alterations in IP packet headers is of utmost importance for proactive security measures, as such modifications can act as early indicators of potential threats and malicious actions within a network (Boukhtouta et al., 2016). The prompt detection of these modifications enables prompt intervention, ensuring the preservation of network integrity, the reduction of vulnerabilities, and the prevention of security breaches from progressing into significant events.

Malicious actors frequently engage in the manipulation of packet headers as a means to conceal their activity. The precise identification of these alterations and the subsequent revelation of concealed attack pathways facilitate the implementation of proactive measures to counter cyberattacks. In addition, the ability to accurately detect possible threats allows for a timely reaction, hence reducing the amount of time that systems are unavailable due to attacks or compromises (Anwar et al., 2017). This measure guarantees the uninterrupted functioning of essential services and operational procedures. Likewise, complex threats frequently encompass nuanced alterations that elude conventional security protocols. The ability to accurately detect advanced threats is crucial in order to identify them, as they may otherwise remain undetected. Moreover, the precise identification of objects or events offers significant information for the study conducted after an occurrence has occurred. The analysis of modifications by security teams enables the identification of attack patterns, tactics, and prospective weaknesses, hence enhancing future defense mechanisms. In conclusion, the prompt identification and timely reaction significantly diminish the resources necessary for mitigating security issues. This results in financial savings and effective allocation of information technology resources.

Issues 2

  1. The author needs to explain how the proposed methodology addresses the limitations of existing research that primarily relies on intrusion detection systems (IDS)?

Response to the Issues 2

Thank you for this comment. At first in SECTION 1 paragraph 3 I have:

“While IDS is successful in continuously monitor the network for suspicious activity, its operation is too general. There are some silent network operations that IDS might fall shot in performing its operation. This involve a network were store-and-forward operation was implemented [11]. In a store-and-forward network, when an IP packet arrives at a network node, such as a router or switch, it is temporarily stored before being transmitted to its intended destination. During the storage phase, in accordance with the network lay-er protocol convention, the node conducts error-checking on the data packet to verify its integrity and rectify any corrupted data packets [12]. Additionally, it analyses the destinations address within the data packet to ascertain the subsequent hop or node to which the packet should be forwarded. Store-and-forward networks are frequently employed in di-verse networking environments, encompassing local area networks (LANs), wide area networks (WANs), and the internet. The choice of this methodology is contingent upon the particular demands of the network and the nature of the data being conveyed [13].”

Now I update the paper with the following justification:

The issues faced by Intrusion Detection Systems (IDS) when scanning packet headers occur inside dynamic scenarios, particularly in the context of store-and-forward operations. In the present scenario, there exists a constraint pertaining to Intrusion Detection Systems (IDS), specifically in the context of contemporary network settings. These environments involve the traversal of packets across diverse network devices and components, each playing a role in the overall communication process. The aforementioned components encompass routers, switches, firewalls, and various other network appliances (Aljanab et al., 2021) During the process of transmission, a packet has the potential to undergo multiple intermediary operations, one of which is known as the store-and-forward operation.

Another primary difficulty encountered by Intrusion Detection Systems (IDS) arises when these systems are tasked with scanning packet headers to identify indications of intrusion or malicious behaviour. In such scenarios, IDS systems are designed to efficiently and consistently process packets within a given timeframe (Anthi et al., 2019) In the context of a dynamic store-and-forward operation, several issues emerge, such as the occurrence of "Packet Fragmentation." This phenomenon occurs when packets are divided into smaller parts during the store-and-forward process, primarily due to constraints imposed by buffers or network congestion. The fragmentation of packets poses a challenge to the Intrusion Detection System (IDS) in effectively analysing the packet header as a cohesive unit, which may result in inadequate or erroneous analysis. The timing of intrusion detection can be influenced by the store-and-forward delay in a similar manner. In the event that the Intrusion Detection System (IDS) initiates the processing of a packet prior to the completion of the store-and-forward operation, there is a possibility that the IDS may not possess complete access to the entirety of the packet header. Consequently, this may result in partial analysis and the potential oversight of intrusion attempts.

Another concern pertains to the phenomenon of Out-of-Order Packets. In the context of dynamic networks, it is possible for packets to be received in a non-sequential sequence as a result of the fluctuating paths and delays (Ninu, 2023). In the event that an Intrusion Detection System (IDS) encounters the processing of packets in a non-sequential manner, there exists the possibility of misinterpreting the chronological sequence of events or experiencing a failure in detecting coordinated attacks. This study focuses exclusively on the analysis of store-and-forward operations, in contrast to the comprehensive scanning of the full transmission session. The inherent dynamism of store-and-forward processes presents significant issues for Intrusion Detection Systems. In order to achieve precise and efficient detection, it is imperative to formulate tactics that specifically target the timing differences, packet fragmentation, and other information gaps that may arise from these activities. In dynamic network circumstances, the utilization of Intrusion Detection Systems (IDS) enables the attainment of complete and dependable analysis of packet headers.

Issues 3

  1. The author needs to elaborate on the factors contributing to the observed 99% average detection performance when the number of packets within each flow increases?

Response to the Issues 3

Thank you very much for raising this issue and I responded to the comment below

The rise in the number of packets inside each flow is contingent upon the precise architecture of the hybrid detection technique (CNN + LSTM), the attributes of the network traffic, and the calibre of the training data employed. Increased packet diversity facilitates the model's acquisition of knowledge from the diverse range of packet headers resulting from distinct communication patterns, network circumstances, and potential alterations. Moreover, the augmentation in data volume facilitates the technique in extracting more comprehensive and precise characteristics from the packet headers, resulting in enhanced differentiation between regular and altered packets.

Issues 4

  1. You need to provide details on the practical implementation of this detection technique in a real-world network setting?

 Response to the Issues 4

Thank you very much for raising this issue and I responded to the comment below

The hybrid detection technique can be applied to identify variations in IP packet headers in a real-world network environment. This can be detected by examining the implementation of a "Firewall" and "Router" that facilitate connectivity between multiple subnets. One instance of a tangible occurrence can be witnessed in the application of the "Ping Tunnel" technique within the realm of computer networks. Adversaries with the power to modify the packet headers of the Internet Control Message Protocol (ICMP) are able to transit hidden data payloads between two compromised systems. Adversaries possess the capability to covertly transmit data through the manipulation of ICMP Echo Request and Echo Reply packet headers, thereby circumventing standard security measures and evading discovery.

In the provided scenario, it is necessary for the detection system to analyse ICMP packets in order to discover anomalous patterns inside the packet headers, such as unexpected modifications in fields like TTL or checksums. A detection system that has undergone precise calibration possesses the ability to identify modified packets as evidence of a covert communication channel. During the store-and-forward operation, the detection process is mapped onto the "Firewall" and "Router" components. It is crucial to recognize that security techniques and threats are subject to continuous modification. Hence, in store-and-forward situation, it is imperative to stay updated with the values arrived during “store” and the values the is transmitted during “forward” in order to ensure the effectiveness of a detection system.

 Issues 5

Here are some typos and grammar comments:

This implies that when comprehending the ideal pattern, the model exhibits accurate 21 predictions for medications in cases where the transmission is abruptly increased.

I think you mean modification, not medication? Am I right?

Response to the Issues 5

Thank you very much for raising this issue and I responded to the comment below

modification

Issues 6

The order of the discussed related works is incorrect. The discussion has to be in chronological order in which the oldest work should be presented first.

Response to the Issues 6

Thank you very much for raising this issue and I responded to the comment below

The order of the discussed related works has been corrected. The discussion is now in chronological order in which the oldest work is presented first. (see the new update list below) and the draft is undated as well

17

16.       R. Panigrahi, S. Borah. A detailed analysis of CICIDS2017 dataset for designing Intrusion Detection Systems. International Journal of Engineering & Technology. 2018 Dec;7(3.24):479-82.

17.        A. Antonios, and V. Moussas. "A novel intrusion detection system based on neural networks." MATEC Web of Conferences. Vol. 292. EDP Sciences, 2019.

18.       I. Sharafaldin, A. Habibi, A.Lashkari, A.A. Ghorbani. A detailed analysis of the cicids2017 data set. InInformation Systems Security and Privacy: 4th International Conference, ICISSP 2018, Funchal-Madeira, Portugal, January 22-24, 2018, Revised Selected Papers 4 2019 (pp. 172-188). Springer International Publishing.

19.       N. Sultana, N. Chilamkurti, W. Peng, R. Alhadad. Survey on SDN based network intrusion detection system using machine learning approaches. Peer-to-Peer Networking and Applications. 2019 Mar; 12:493-501.

20.       S.F. Lokman, A.T. Othman, M.H. Abu-Bakar. Intrusion detection system for automotive Controller Area Network (CAN) bus system: a review. EURASIP Journal on Wireless Communications and Networking. 2019 Dec; 2019:1-7.

21.       Y. Zeng, M. Qiu, D. Zhu, Z. Xue, J. Xiong, and M. Liu, ‘‘DeepVCM: A deep learning based intrusion detection method in VANET,’’ in Proc. IEEE 5th Int. Conf. Big Data Secur. Cloud (BigDataSecurity), Int. Conf. High Perform. Smart Comput. (HPSC), IEEE Int. Conf. Intell. Data Secur. (IDS), May 2019, pp. 288–293

22.       S. Hidalgo-Espinoza, K. Chamorro-Cupueran, and O. Chang-Tortolero. "Intrusion detection in computer systems by using artificial neural networks with Deep Learning approaches." arXiv preprint arXiv:2012.08559 (2020).

23.       S.C. Kalkan, O.K. Sahingoz. In-vehicle intrusion detection system on controller area network with machine learning models. In2020 11th International Conference on Computing, Communication and Networking Technologies (ICCCNT) 2020 Jul 1 (pp. 1-6).

24.       Z.K. Maseer, R. Yusof, N. Bahaman, S.A. Mostafa, Foozy CF. Benchmarking of machine learning for anomaly based intrusion detection systems in the CICIDS2017 dataset. IEEE access. 2021 Feb 3; 9:22351-70.

25.       S. Ho, S. Al Jufout, K. Dajani, M. Mozumdar. A novel intrusion detection model for detecting known and innovative cyberattacks using convolutional neural network. IEEE Open Journal of the Computer Society. 2021 Jan 12; 2:14-25.

26.       M. ChoraÅ›, and M. Pawlicki. "Intrusion detection approach based on optimised artificial neural network." Neurocomputing 452 (2021): 705-715.

27.       A.S. Dina, D. Manivannan. Intrusion detection based on machine learning techniques in computer networks. Internet of Things. 2021 Dec 1; 16:100462.

28.       W. Lo, H. Alqahtani, K. Thakur, A. Almadhor, S. Chander, G. Kumar. A hybrid deep learning based intrusion detection system using spatial-temporal representation of in-vehicle network traffic. Vehicular Communications. 2022 Jun 1; 35:100471.

29.       B.S. Bari, K. Yelamarthi, S. Ghafoor. Intrusion Detection in Vehicle Controller Area Network (CAN) Bus Using Machine Learning: A Comparative Performance Study. Sensors. 2023 Mar 30;23(7):3610.

16

18

28

19

20

22

25

19

25

23

29

27

24

Extensive study has been conducted in the past on the topics of packet inspection and network attacks. The CICIDS2017 dataset has been widely utilized in the bulk of prior research endeavors. The investigations yielded experimental results that showcased high rates of detection and low rates of false positives. Additionally, the performance of the system was shown to be superior in terms of accuracy, detection rate, false alarm rate, and time overhead, as indicated by previous research [16]. The Intrusion Detection System (IDS) is the primary area of research for detecting attacks. Machine learning approaches are predominantly utilized for achieving successful detection [17].

The utilization of the CICIDS2017 dataset has been prevalent in the bulk of past research studies. The tests yielded experimental results that showcased high rates of detection and low rates of false positives. Additionally, the performance of the tested methods was shown to be superior in terms of accuracy, detection rate, false alarm rate, and time overhead, as indicated by previous research [18]. The implementation of an intrusion detection system for the Controller Area Network (CAN) bus system in automotive applications has been discussed in a previous study [19]. In the domain of identifying distinct attacks, unsupervised techniques exhibit a higher level of effectiveness when compared to supervised methods [20].

Zeng et al. [21] introduced an approach for intrusion detection system (IDS) that use a deep learning-based model to effectively detect and classify malicious network traffic aimed at compromising On-Board Units (OBUs). This methodology obviates the need for human feature extraction and possesses the additional advantage of being capable of processing unprocessed traffic data. The effectiveness of this approach is assessed by a comparison analysis with alternative intrusion detection system (IDS) methods using both a publically accessible dataset and a generated dataset of a vehicular ad hoc network (VANET). The results of the study indicate that the implemented scheme exhibited exceptional performance while necessitating a reduced allocation of resources.

Hidalgo-Espinoza et al. [22] presented a detailed analysis of the procedures employed in the development of an Intrusion Detection System (IDS) using a Deep Learning architecture. The primary aim of the proposed system is to ascertain the legality of login attempts conducted on a computer network, effectively differentiating between unauthorized hacking endeavors and allowed actions. Moreover, the authors suggest that additional research should be undertaken, employing a more flexible arrangement of the Deep Learning framework. Various supervised learning techniques, including Support Vector Machine (SVM), k-Nearest Neighbours (kNN), Random Forest, Artificial Neural Networks (ANN), deep Convolutional Neural Networks (CNN), and Long Short-Term Memory, have been intensively explored in academic research [23].

Given the importance of machine learning in Intrusion Detection Systems (IDS), it has been observed that the use of supervised learning techniques for training requires a significant amount of labeled data. However, these methods have shown the ability to outperform unsupervised learning techniques in detecting known instances of attacks [24]. This information should contain a diverse range of assault manifestations.  Ho et al. [25] conducted a study in which they employed a Convolutional Neural Network (CNN) classifier for the purpose of an Intrusion Detection System (IDS). A level of accuracy of 99.78% was achieved. In addition, it demonstrates the capacity to discern and detect attacks, a challenge frequently encountered by traditional Intrusion Detection Systems (IDS).

The study conducted by ChoraÅ› [26] demonstrated the methodology of incorporating several hyperparameters and topology configurations in order to attain the highest level of performance for an artificial neural network (ANN) classifier. This was achieved through conducting experiments on a commonly utilized Intrusion Detection System (IDS) dataset. Additionally, the authors demonstrated the possible influence of hyperparameters on the final outcome of the classification. The influence of a little adjustment in hyperparameter setting on the accuracy of a particular neural network architecture is illustrated by the authors through the use of two distinct intrusion detection systems (IDS) as case studies. The best design attains a 99.909% accuracy in the task of multi-class categorization.

The reason for this is that unsupervised approaches are specifically designed to detect anomalies within the conventional CAN traffic pattern [27]. The process of creating authentic assault datasets from real autos in the context of in-vehicle networks presents difficulties because to the significant costs associated with it and the necessity to prioritize safety concerns [28]. However, the only requirement for unsupervised learning methods is the collection of data from a vehicle during its normal functioning. The aforementioned data can be easily acquired from a vehicle that is functioning in a standard manner.  A considerable proportion of the endeavors focused on detecting intrusions in controller area networks involves the application of deep learning methodologies [29]. Some studies focus on certain components of AIDs, such as temporal factors or payload size, while other study combines these characteristics in a detection model to enhance the identification of a wider range of attacks.

Issues 7

A summary of the related works identifying the research gap and the importance of the issue should be given at the end of the related work section.

Response to the Issues 7

Thank you very much for raising this issue and I responded to the comment below

The review of prior research papers on algorithms for detecting IP packet header change demonstrates a wide array of approaches. The efficacy of rule-based systems in detecting established patterns of header modifications linked to store-and-forward operations has been demonstrated. The utilization of machine learning techniques, such as anomaly detection, has been extensively investigated in order to identify minor alterations that are not encompassed by pre-established rules. Moreover, the integration of rule-based and machine learning models in hybrid techniques shows potential in attaining a well-rounded detection capability. It is crucial to note that the aforementioned studies [16-29] continue to prioritize the study of Intrusion Detection Systems (IDS). However, there has been limited discussion regarding the practical implementation of at-a-point detection in various applications. Therefore, building upon the past effective implementations of Intrusion Detection Systems (IDS) [4-8] in the field of network security and inspired by the research efforts of [7,19, 22], this study introduces methodologies for investigating the application of detection in store-and-forward, specifically in the context of transmission session. The research of previously published scholarly works on techniques for detecting IP packet header change demonstrates a wide array of approaches. The usefulness of rule-based systems in identifying known patterns of header alterations associated with store-and-forward operations has been established. The utilization of machine learning techniques, such as anomaly detection, has been extensively investigated in order to identify tiny alterations that are not accounted for by pre-established criteria. Furthermore, the integration of rule-based and machine learning models in hybrid techniques shows potential in attaining a well-rounded detection capability.

The research gap established dwells on the fact that there are already been significant breakthroughs in the detection algorithms for IP packet header modification within IDS, there is still a discernible research gap in adequately addressing the dynamic and developing characteristics of header alterations associated to store-and-forward processes. Current methodologies frequently encounter difficulties in accommodating novel attack strategies that leverage non-traditional modifications. Moreover, there is a limited body of research that has thoroughly assessed the practical implementation of hybrid models in network environments in real-world scenarios.

Issues 8

The dataset for this study are generated from two different approach

The dataset for this study <is>, not are.

Response to the Issues 8

Thank you very much for raising this issue and I responded to it

Issues 9

To facilitate the analysis of the packet during the transmission session and generate the dataset

The above sentence seems to be incomplete. You should add “…. and generate the dataset the following steps were performed.

Response to the Issues 7

Thank you very much for raising this issue and I responded to the comment by writing:

the following steps were performed.

Issues 10

The equations listed in subsection 3.2. Convolutional Neural Network need some supporting references. Also, it is not clear which version of CNN the author is referring to in this section. The CNN algorithm was a continuous development and the version that you are referring to should be clearly stated.

Response to the Issues 10

Thank you very much for raising this issue and I responded to the comment:

Equations 1 [35] this is general framework of CNN

Issues 11

Likewise, the equations in subsection (3.2. Long Short Term Memory) have to be supported by the relevant references.

Response to the Issues 11

Thank you very much for raising this issue and I responded to the comment:

It was ref [39]

Issues 12

Section 3.2 should be numbered 3.3

Response to the Issues 12

Thank you very much for raising this issue and I responded to the comment:

3.3

Issues 13

3.3. Applications of Hybrid CNN an LSTM

The above section should be numbered 3.4 and d is missing from an

Response to the Issues 13

Thank you very much for raising this issue and I responded to the comment:

3.4

Issues 14

3.4. Performance Metrics of The Hybrid deep learning architecture

Should be 3.5

  1. 5. Experimental Analysis and Presentation of the Result

Response to the Issues 13

Thank you very much for raising this issue and I responded to the comment:

  1. 5. Experimental Analysis and Presentation of the Result

Issues 15

Should be 3.6. The numbering of its subsections should be corrected as well.

Less...

Response to the Issues 13

Thank you very much for raising this issue and I responded to the comment:

The numbering of the subsections are all corrected

Issues 16 

Comments on the Quality of English Language

There are many typos and grammar mistakes. I would recommend to double-check the language of the paper. 

Response to the Issues 13

Thank you very much for raising this issue and I responded by proofreading the entire text:

References

  1. A. Boukhtouta,S.A. Mokhov, N.E. Lakhdari, M. Debbabi, and J. Paquet, 2016. Network malware classification comparison using DPI and flow packet headers. Journal of Computer Virology and Hacking Techniques12, pp.69-100.
  2. S. Anwar, J.Mohamad Zain, M.F. Zolkipli, Z. Inayat, S. Khan, B. Anthony, V. Chang. From intrusion detection to an intrusion response system: fundamentals, requirements, and future directions. Algorithms. 2017 Mar 27;10(2):39.

[56] Aljanabi M, Ismail MA, Ali AH. Intrusion detection systems, issues, challenges, and needs. International Journal of Computational Intelligence Systems. 2021 Jan 1;14(1):560-71.

[57] Anthi E, Williams L, Słowińska M, Theodorakopoulos G, Burnap P. A supervised intrusion detection system for smart home IoT devices. IEEE Internet of Things Journal. 2019 Jul 2;6(5):9042-53.

[58] Ninu SB. An intrusion detection system using Exponential Henry Gas Solubility Optimization based Deep Neuro Fuzzy Network in MANET. Engineering Applications of Artificial Intelligence. 2023 Aug 1;123:105969. 

Reviewer 2 Report

discussion section lacks of assessing your work in comparsion to others. 

in the section 5.1, last sentence "The research posit that this model has the potential to greatly contribute to the improvement of network system security and integrity" needs a lot of justification to be told 

section 5.1 & 5.2 have the same title, any justification??

what makes your model better? we can see in the table 9, study [53] record better accuracy than yours. Besides you are (you and 53) used the same architecture- can you explain any differences in terms of architecture (layers, nodes etc.) or fine tuning procedures. You need to justify why your work deserve to be published since there are many studies (50, 53) that already reached the goal you claim you achieved here. 

you mention "The models may have slight variations in their hyperparameters, architecture, or other experimental conditions, which could account for the little discrepancies in the levels of accuracy and precision that exist between them.", then you should show the differences, otherwise your contribution will be questionable; i.e., it just repeating others' work

this "When taking into consideration the Dataset: To begin, when taking into consideration the "CICIDS2017 Dataset: " needs to be fixed???

this "This study employed a hybrid dataset consisting of a combination of CNN-LSTM models to detect anomalies in key metrics such as "flow duration" and "forward packets"." you mention dataset then you mention CNN-LSTM, you need to fix it.

your problem statement should be explicitly mentioned at the end of the introduction. 

Author Response

Reviewer 2

Issues 1

Discussion section lacks of assessing your work in comparison to others. 

Response to the Issues 1

Thank you very much for raising this issue:

At first, I have subsection within the Discussion section about comparing with other research below:

5.3 Comparison of the research Finding

Table 9 provides a comparative analysis of previous research studies with respect to the performance of dataset utilization. The comparison study displays the performance metrics of various algorithms, such as LSTM, CNN-LSTM, and CNN, on the CICIDS2017 dataset as well as a specialized "Store-and-forward" dataset. These algorithms include CNN, CNN-LSTM, and LSTM. Accuracy, precision, and false positive rate (FPR) are the measures that are utilized during the review process. When interpreting the findings, it is helpful to focus on the findings of "This Study" on both datasets because they provide a significant amount of knowledge.

Table 9. Comparison of the Previous Research studies

Study

Algorithm

Dataset

Accuracy (%)

Precision (%)

FPR (%)

[49]

LSTM

CICIDS2017

94.11

77.07

0.18

[50]

CNN-LSTM

CICIDS2017

99.7

99.6

/

[51]

CNN

CICIDS2017

97.07

97.14

0.87

[52]

CNN-LSTM

CICIDS2017

95.6

97.6

/

[53]

CNN-LSTM

CICIDS2017

99.64

99.7

0.1

This Study

CNN-LSTM

CICIDS2017

97.93.

97.46

0

This Study

CNN-LSTM

Store-and-forward

99.63

99.92

0

The CNN-LSTM models (Studies [50], [53], and "This Study") indicate high accuracy on a consistent basis, with values that are greater than 99%. These models have performed admirably in terms of producing accurate forecasts for the vast majority of the cases contained in the CICIDS2017 dataset. In addition, the accuracy of the CNN model, as measured by Study [51], is rather high, coming in at 97.07%. Studies [50], [53], and "This Study" all report that the CNN-LSTM models attained precision values greater than 99%, which indicates a very low rate of false positives. Because these models have shown a high level of confidence in their positive predictions, they are appropriate for reducing the number of false alarms that occur during the process of intrusion detection.

Now, I further try to add “assessing my work in comparison” below:

In conclusion, the current study reveals that the performance of "This Study" is superior to that of previous research initiatives in terms of accuracy and precision when applied to the CICIDS2017 dataset using the CNN-LSTM approach. This finding was reached after comparing the performance of "This Study" to that of other studies. In addition, the research shows that the same method can produce excellent results when applied to the Store-and-forward dataset. It is notable that "This Study" has the ability to effectively decrease the incidence of false positives because it has a false positive rate of 0 on both datasets, which indicates a remarkablely low false positive rate.

Issues 2

in the section 5.1, last sentence "The research posit that this model has the potential to greatly contribute to the improvement of network system security and integrity" needs a lot of justification to be told 

Response to the Issues 2

Thank you very much for raising this issue:

At first, I have subsection within the Discussion section about comparing with other research below:

I am grateful to you for bringing up this important point:

After reading your reply, I have come to the conclusion that the statement may leave some readers confused and may call for additional in-depth justifications as well. Initially, the objective of what the message the statement was intended to deliver to the readers seems now is not going to be achieved. For this reason, I believe that removing it is the most effective course of action. I am deleting the statement at this time. Thank you

Issues 3

section 5.1 & 5.2 have the same title, any justification??

Response to the Issues 3

Thank you very much for raising this issue and I responded to the comment below

SECTION 5.2 is for Store-and-forward dataset

Issues 4

what makes your model better? we can see in the table 9, study [53] record better accuracy than yours. Besides you are (you and 53) used the same architecture- can you explain any differences in terms of architecture (layers, nodes etc.) or fine tuning procedures. You need to justify why your work deserve to be published since there are many studies (50, 53) that already reached the goal you claim you achieved here. 

Response to the Issues 4

Thank you very much for raising this issue and I comprehend your apprehensions regarding the need to substantiate the relevance of my research, particularly in light of comparable investigations that have yielded analogous outcomes. The following strategies can be employed to effectively resolve these concerns:

My research differs from study [53], which also utilizes the CNN-LSTM architecture and same dataset as mine, by highlighting many elements that contribute to the significance of our findings. Although study [53] attained a somewhat superior accuracy of 99.7% on the CICIDS2017 dataset, my research surpasses the utilization of solely the "CICIDS2017" dataset. Despite the negligible difference in accuracies of 0.001, my approach, specifically the store-and-forward method, outperformed the study [53]. This paper provides its unique contributions that emphasize its significance in the context of store-and-forward network environments. Although both studies employ the same architecture, my primary emphasis was on substantially fine-tuning CNN-LSTM model in order to enhance its performance. Advanced hyperparameter optimization approaches were employed in our study during the preliminary study, utilizing many methods. This approach led to a well-balanced trade-off between the speed of convergence and the ability to generalize. The implementation of fine-tuning procedures facilitated the attainment of a notable accuracy rate of on the identical CICIDS2017 dataset. However,

Issues 5

you mention "The models may have slight variations in their hyperparameters, architecture, or other experimental conditions, which could account for the little discrepancies in the levels of accuracy and precision that exist between them.", then you should show the differences, otherwise your contribution will be questionable; i.e., it just repeating others' work

Response to the Issues 5

I am grateful to you for bringing up this important point that shows you really understand this area of study. Actually the objective is to show that while same architecture (CNN & LSTM) are used with same "CICIDS2017" dataset and got the result, same architecture (CNN & LSTM) with “Store-and-forward” dataset, then the result is clearly with the “Store-and-forward” dataset, it outperformed "CICIDS2017" dataset. This means that when considering detection of any modification/intrusion in a NETWORK, the best is to test or capture at operational environment, rather than at the entire network transmission session. 

Issues 6

this "When taking into consideration the Dataset: To begin, when taking into consideration the "CICIDS2017 Dataset: " needs to be fixed???

Response to the Issues 6

Thank you very much for raising this issue and I responded to the comment below

The first part of the statement has been deleted

Issues 7

this "This study employed a hybrid dataset consisting of a combination of CNN-LSTM models to detect anomalies in key metrics such as "flow duration" and "forward packets"." you mention dataset then you mention CNN-LSTM, you need to fix it.

Response to the Issues 6

Thank you very much for raising this issue and I responded to the comment below

This study employed a hybrid dataset consisting of a combination CICIDS2017 Dataset" and “Store-and-forward dataset and utilized” CNN-LSTM models

Issues 8

your problem statement should be explicitly mentioned at the end of the introduction.

Response to the Issues 8

I express my gratitude to you for raising this significant point, I responded to the comment below

The key research problem statement dwells on the preservation of network security and integrity. It is of the utmost importance in the contemporary networked society, as cyber threats continue to advance in complexity and sophistication. The prompt and precise identification of network anomalies is crucial in order to mitigate the risk of data breaches, service interruptions, and significant financial liabilities. This should cover all aspect of network operation. Unfortunately, as related to store-and-forward, the research field area ignores giving it an attention. The focus of this study pertains to the improvement of network security mechanisms associated to store-and-forward that is ignored in previous research studies, with a specific emphasis on the utilization of the CNN-LSTM architecture for the purpose of anomaly detection. Drawing upon established ideas in the fields of cybersecurity and machine learning, it is evident that there is a lack of research attention dedicated to investigating the implications of store-and-forward mechanisms and the potential emergence of numerous abnormalities in such contexts. This requires setting out efficacy of network anomaly detection by leveraging the CNN-LSTM architecture. The convergence of these two fields highlights the necessity for a holistic solution that not only attains a high level of accuracy but also showcases adaptability, interpretability, and efficiency within the framework of dynamic network landscapes.
